# Training Prompt Matters: State-Adaptive Optimization for Robust Fine-Tuning

**Wenhang Shi** [1]  **Yiren Chen** [2]  **Shuqing Bian** [3]  **Zhe Zhao** [3]  **Jinhao Dong** [1]  **Pengfei Hu** [3]  **Wei Lu** [1]  **Xiaoyong Du** [1]

## Abstract

While prompt engineering is instrumental in maximizing the capabilities of Large Language Models (LLMs) during inference, the role of prompts during training remains critically underexplored. Prevailing fine-tuning paradigms typically treat training prompts as mere surface forms, assuming that semantically equivalent instructions yield identical learning outcomes. However, we reveal that this equivalence is deceptive: while paraphrased prompts often lead to comparable in-task performance, they induce drastically different cross-task impacts regarding catastrophic forgetting and generalization. Crucially, these impacts are positively correlated across tasks, indicating the existence of superior prompts that consistently yield better performance. Furthermore, we discover that these superior prompts can be robustly identified by task loss prior to learning. Leveraging these insights, we introduce State-Adaptive Prompt Optimization (SAPO), a lightweight yet effective training strategy that shifts task formulation from a static input to a dynamic, state-adaptive variable. Comprehensive experiments on diverse benchmarks confirm its effectiveness, which significantly mitigates forgetting while improving generalization, achieving substantial performance gains over state-of-the-art methods. These results provide insights into how training prompts shape learning dynamics and offer a practical recipe for robust fine-tuning. Our code is available at https://github.com/Eric8932/SAPO.

## 1. Introduction

Large Language Models (LLMs) exhibit pronounced sensitivity to prompt design during inference, where even minor variations can drastically alter task-solving behaviors and performances (Liu et al., 2023; Dubey et al., 2024; Achiam et al., 2023). Consequently, prompt engineering has become a standard practice for maximizing LLM capabilities in specific tasks (Zhou et al., 2023; Sahoo et al., 2024). However, while the impact of prompts during inference is well-studied, their role in the construction of training data for fine-tuning remains critically underexplored. In prevailing paradigms, training prompts are typically treated as static, arbitrary choices, operating under the assumption that semantically equivalent instructions yield identical learning outcomes (Wang et al., 2022; Yue et al., 2023; Luo et al., 2025).

Contrary to this typical view, our study reveals that such semantical equivalence is deceptive. When models are trained using different paraphrased prompts for the same task, their in-task performance remains largely consistent, which explains why training prompt engineering is often overlooked. However, a radically different picture emerges when examining the model's broader capabilities. The choice of training prompt exerts a profound impact on catastrophic forgetting of previously learned tasks and generalization to unseen tasks (McCloskey & Cohen, 1989; Brown et al., 2020), leading to divergent cross-task behaviors even among semantically indistinguishable prompts. Crucially, these variations are not random, but exhibit a consistent alignment where training prompts that mitigate forgetting also tend to facilitate generalization. This positive correlation across tasks implies the existence of superior training prompts, rendering prompt formulation a tractable optimization objective.

Given the necessity and feasibility of training prompt engineering, the challenge shifts to efficiently identifying the superior prompts prior to learning. Following established works that compute statistical correlations to identify performance predictors, we conduct a comprehensive investigation of potential indicators (Lin, 2004; Radford et al., 2019; Sun et al., 2025). We discover that the superior prompts can be robustly identified via **pre-update loss**. Specifically, prompts with lower loss consistently mitigate forgetting and enhance generalization. Leveraging these insights, we propose State-Adaptive Prompt Optimization (SAPO), a lightweight yet effective training strategy that shifts task formulation from a static input to a dynamic, state-adaptive variable. Instead of utilizing fixed training data, SAPO actively aligns prompts with the model's evolving state. Specifically, before learning a task, SAPO generates

[1]School of Information, Renmin University of China, Beijing, China [2]Peking University, Beijing, China [3]Tencent , Beijing, China. Correspondence to: Jinhao Dong <dongjinhao@ruc.edu.cn>.

multiple paraphrased candidates, evaluates their alignment to model's current state using pre-update loss, and integrates the optimal prompt for training. By better leveraging model's intrinsic capabilities through lower-loss training prompts, SAPO minimizes the disruptive, task-specific adaptations that interfere with other tasks, thereby facilitating generalizable knowledge acquisition.

Due to its focus on input alignment, SAPO is orthogonal to existing training strategies and allows for seamless integration to transform fixed, state-agnostic training processes into state-adaptive ones. Comprehensive evaluations on diverse benchmarks confirm that SAPO achieves substantial performance gains over state-of-the-art methods, effectively reducing forgetting while improving zero-shot generalization. Our contributions are summarized as follows:

- **Systematic study of training prompt impact.** We provide the first systematic study of the role of training prompts in LLM fine-tuning, revealing that while semantically equivalent prompts have negligible impact on the current task, they are critical factors in determining the model's cross-task capabilities, including forgetting and generalization.

- **Existence and identification of superior prompts.** We demonstrate the existence of superior training prompts and show they are identifiable via pre-learning loss. This establishes a predictive link between the model's current state and optimal task formulation.

- **State-Adaptive Prompt Optimization (SAPO) method.** We propose a lightweight, plug-and-play training strategy that dynamically optimizes prompts based on model's state before fine-tuning. SAPO achieves significant and robust performance gains over baselines across various models and tasks.

## 2. Related Work

### 2.1. Prompt Engineering

LLMs exhibit high sensitivity to prompt design: evaluation performance can fluctuate sharply with even minor variations in task instructions (Liu et al., 2023; Achiam et al., 2023; Dubey et al., 2024). Even input perturbations, which remain transparent to human comprehension, can induce substantial shifts in model outputs (Zhan et al., 2024). Consequently, prompt engineering is crucial for adapting LLMs to downstream tasks (Sahoo et al., 2024). To automate this process, prior works use reinforcement learning to compose prompt tokens or employ LLMs to iteratively refine prompts (Zhang et al., 2023; Kong et al., 2025; Zhou et al., 2023; Shi et al., 2025). However, they predominantly focus on inference-time usage. In this study, we present the first systematic investigation into the role of training prompts, showing their profound cross-task impacts and the existence

of identifiable superior prompts, underscoring the necessity and feasibility of training prompt engineering.

### 2.2. Forgetting and Generalization in Fine-Tuned LLMs

Adapting LLMs to specific tasks via fine-tuning often degrades their broad capabilities, most notably causing catastrophic forgetting on trained tasks and diminished generalization to unseen ones (McCloskey & Cohen, 1989; Luo et al., 2023; Zhang & Wu, 2024; Wu et al., 2024). Existing remedies from the continual learning domain generally fall into three families: (i) regularization of parameter updates (Kirkpatrick et al., 2016; Huang et al., 2021), (ii) replay of prior or self-synthesized data (Scialom et al., 2022; Huang et al., 2024; Wang et al., 2024), and (iii) modularization with task-specific adapters (Wang et al., 2023a; Razdaibiedina et al., 2023; Wang et al., 2023c). However, these approaches typically apply fixed task formulations irrespective of model's continuously evolving state. In this work, we introduce adaptive prompt optimization, which actively optimizes prompts based on model's current state before each task, aligning the training context with model's ongoing learning dynamics. While recent reinforcement learning methods similarly emphasize the importance of dynamic data construction (Lu & Lab, 2025; Chen et al., 2025; Mukherjee et al., 2025), they focus on sampling on-policy outputs rather than adaptively optimizing input formulation.

### 2.3. Analysis of Fine-Tuning Mechanisms

Prior work examines how LLMs acquire new abilities during fine-tuning (Ferrando et al., 2024; Wang et al., 2025), ranging from learning minimal wrappers atop existing abilities (Jain et al., 2023) to enhance established capabilities acquired during pre-training (Ren et al., 2024; Prakash et al., 2024). Recent studies decompose task solving into input activating function and intrinsic ability. They discover that fine-tuning primarily modulates input activation pathways rather than creating new capabilities, and performance shifts on other tasks arise from conflicts in activation pathways rather than the destructive overwriting of task-processing functions (Kotha et al., 2024; Zheng et al., 2025; Jiang et al., 2025). Our work advances this understanding by revealing the critical yet overlooked role of training prompts. By strategically varying prompts, one can identify pathways with minimal conflicts, thereby mitigating cross-task performance drifts. This highlights that training prompt engineering is not merely a surface-level adjustment, but an effective strategy for managing interference during fine-tuning.

## 3. The Impact of Training Prompts

To investigate the necessity of training prompt engineering, we examine the research question: *Does the choice of training prompt matter when fine-tuning LLMs, and if*

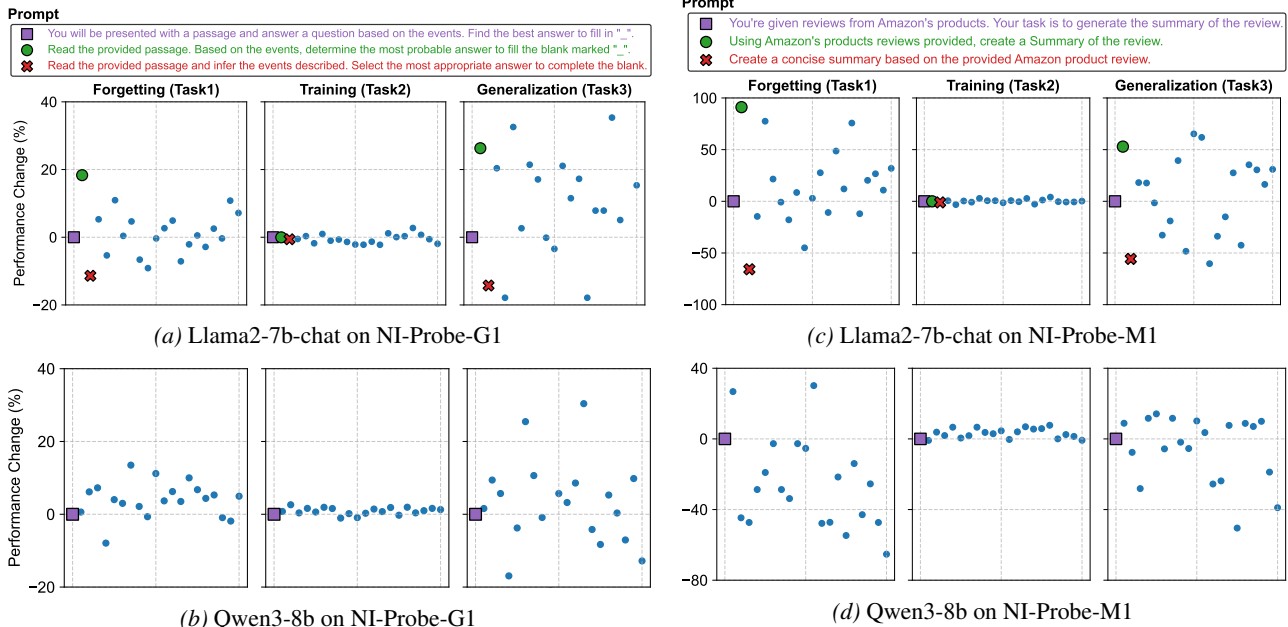

*Figure 1.* Normalized relative performance change (vs. the original prompt) on the trained, current, and unseen tasks after training with paraphrased prompts. Results are for two sequences on Llama-2-7b-chat and Qwen3-8b models. ■ marks the original prompt.

*so, how does it influence model capabilities?* This section presents a systematic study on the effects of fine-tuning with semantically equivalent prompts.

### 3.1. Settings

Given a language model $M$, we train and evaluate it on a three-task sequence $(T_1, T_2, T_3)$, representing a previously trained task, the current target task, and an unseen task, respectively. Each task is associated with a human-crafted prompt $(P_1^0, P_2^0, P_3^0)$. For the current task $T_2$, we additionally generate 20 paraphrased prompts $\{P_2^j\}_{j=1}^{20}$, ensuring they match the semantics and length of the original $P_2^0$ (as shown in the top of Figure 1). All prompts follow a consistent format of simple sentences describing the task execution. The tasks are drawn from SuperNI (Wang et al., 2022), a collection of NLP tasks with expert-written instructions. This benchmark is widely adopted to assess cross-task conflicts and generalization following model fine-tuning (Jiang et al., 2025; Feng et al., 2025). As detailed in Table 5, our evaluation involves 26 diverse datasets covering a broad spectrum of capabilities, including question answering, summarization, and program execution. For clarity, we broadly categorize these datasets into classification and generation tasks. Crucially, these tasks are unseen to the pre-trained model $M$, making this an ideal testbed to systematically investigate the impact of training prompts.

We attempt to quantify how semantically indistinguishable prompts impact model performance across tasks. First, we fine-tune $M$ on $T_1$ with $P_1^0$ and obtain $M_1$. Next, we fine-

tune $M_1$ on $T_2$ using one of the $\{P_2^j\}_{j=0}^{20}$, yielding 21 fine-tuned variants $\{M_2^j\}_{j=0}^{20}$. Finally, we evaluate each variant $M_2^j$ on: (1) $T_1$ (forgetting evaluation) using $P_1^0$; (2) $T_2$ (in-task evaluation) using its respective training prompt $P_2^j$; (3) $T_3$ (generalization evaluation) using $P_3^0$. Detailed protocols are in § 5.1. To verify the universality of our findings, we evaluate across varying model families (Llama and Qwen), scales (7b, 8b, 14b) and task sequence types, generation-only (NI-Probe-G), classification-only (NI-Probe-C), and mixed (NI-Probe-M) sequences. Full probe dataset construction details are available in Appendix A.

### 3.2. Divergent Cross-Task Impacts

Figure 1 illustrates the impact of training prompts for Llama-2-7b-chat and Qwen3-8b models (Touvron et al., 2023; Yang et al., 2025) on generation and mixed task sequences, NI-Probe-G1 and NI-Probe-M1. Each data point corresponds to a paraphrased training prompt for the current task. The y-axis reports the normalized relative performance change, defined as $(S_{\text{variant}} - S_{\text{original}})/S_{\text{original}} \times 100\%$, where $S$ denotes the performance score on the evaluation metric.

**Observation 1: In-task Stability vs. Cross-task Sensitivity.** As shown in Figure 1 middle panels, the performances for all prompt variants nearly coincide, indicating paraphrased prompts have negligible impact on current task performance. In contrast, the side panels show that different prompt choices induce drastic variability in both forgetting (on $T_1$) and generalization (to $T_3$). For example, on NI-Probe-M1 with Llama-2-7b-chat (Figure 1c), the largest

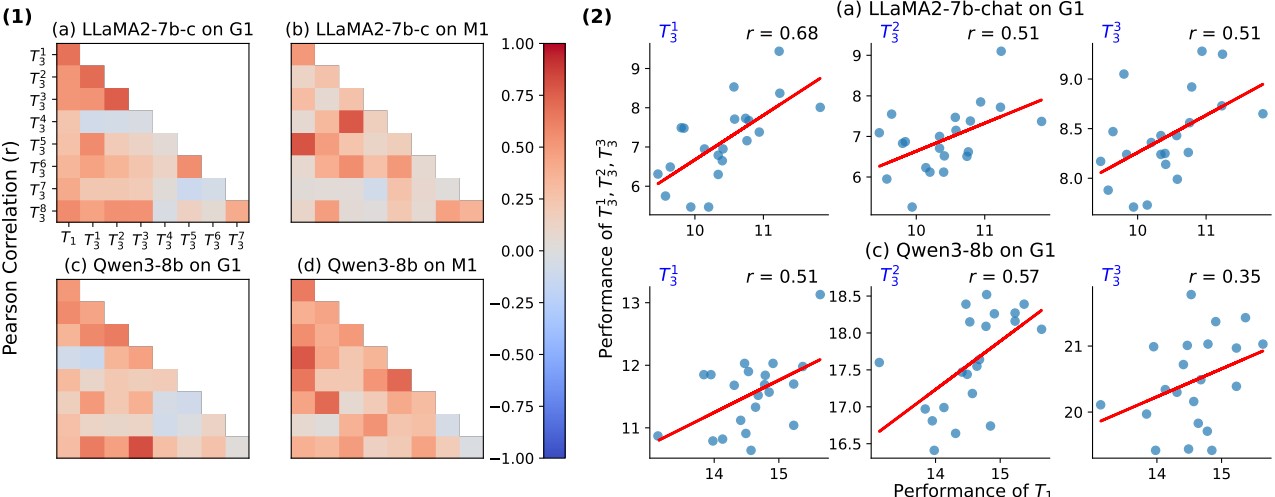

*Figure 2.* (1) Heatmaps of pairwise Pearson correlations among performances across the trained task $T_1$ and eight unseen tasks $\{T_3^j\}$. Each subplot shows a combination between Llama2-7b-chat/Qwen3-8b model and a generative/mixed sequence. (2) Example scatter plots for some task pairs, with x- and y-axis showing performance on the trained and unseen tasks, respectively.

difference across paraphrases reaches 156% for forgetting and 110% for generalization. Moreover, relative to the original instruction, certain paraphrases can simultaneously mitigate forgetting and enhance generalization. For example, in Figure 1c, changing prompt form "*Create a concise summary based on the provided Amazon product review*" ($\times$) to "*Using Amazon's products reviews provided, create a Summary of the review*" ($\bullet$) improves trained and unseen performance from -65/-55% to +91/+52%. Crucially, these prompts differ only in minor lexical and syntactic choices. Yet, even such minimal variations drive the model into vastly different states, suggesting that the impact of prompt formulation is likely even more pronounced for more complex or semantically diverse prompts. These observations are robust across diverse settings, with results for additional sequence types and models in Appendix A (Figures 5 and 6) exhibiting trends strictly consistent with Figure 1. Therefore, the choice of training prompt matters: it is critical in shaping the model's broader capability, impacting the extent of both forgetting and generalization.

### 3.3. Existence of Superior Training Prompts

The observation that specific prompts can simultaneously enhance performance on trained and unseen tasks suggests that these effects are not stochastic. We therefore investigate whether this cross-task impact is systematic, specifically seeking to analyze the existence of universally superior prompts that consistently benefit diverse non-training tasks. We expand our study to encompass 120 task sequences. For each of three sequence categories (generation, classification, mixed), we instantiate five distinct training sequences ($T_1$ and $T_2$) and enlarge the unseen evaluation tasks to eight choices ($\{T_3^j\}_{j=1}^8$). Full construction details appear in Ap-

pendix A. For each sequence, 21 model variants, trained on $T_1$ with $P_1$ and $T_2$ with 21 distinct prompts ($\{P_2^j\}_{j=0}^{20}$), are evaluated on nine non-current tasks ($T_1$ and $\{T_3^j\}_{j=1}^8$). We then compute Pearson correlation (Pearson, 1894) of performance scores across these 21 variants for every pair of evaluation tasks. Figure 2 visualizes these relationships. Panel (1) displays pairwise correlation heatmaps for four model–sequence pairs, where each cell quantifies correlation across 21 prompt variants between two specific tasks. Panel (2) provides a granular view of the performance relationship between $T_1$ (x-axis) and $\{T_3^j\}_{j=1}^3$ (y-axis), with each point representing a specific prompt variant.

**Observation 2: Consistent Performance Coupling.** Panel (2) shows clear positive correlations: prompts that mitigate forgetting on $T_1$ typically yield better performance on $T_3$. This trend is further corroborated on a global scale by Panel (1), where the heatmaps display widespread strong positive correlations (often up to 0.6), indicating a tight performance coupling of prompt effects. Crucially, while minor negative correlations exist, likely because the evaluation task is loosely related to the training task, the dominant trend is positive. This implies that a prompt beneficial for one non-training task is likely to confer benefits to others. Therefore, there exist superior training prompts that consistently improve cross-task performance. The robustness of these findings are verified across diverse settings, including varying sequence types, larger model architectures, and alternative correlation assessments (e.g., the Spearman coefficient (Reimers et al., 2016)). Comprehensive additional results (Figure 7–10) are detailed in Appendix B.2. Consequently, the significance of training prompts extends beyond inducing drastic performance variability; their systematic consistency renders prompt formulation a tractable

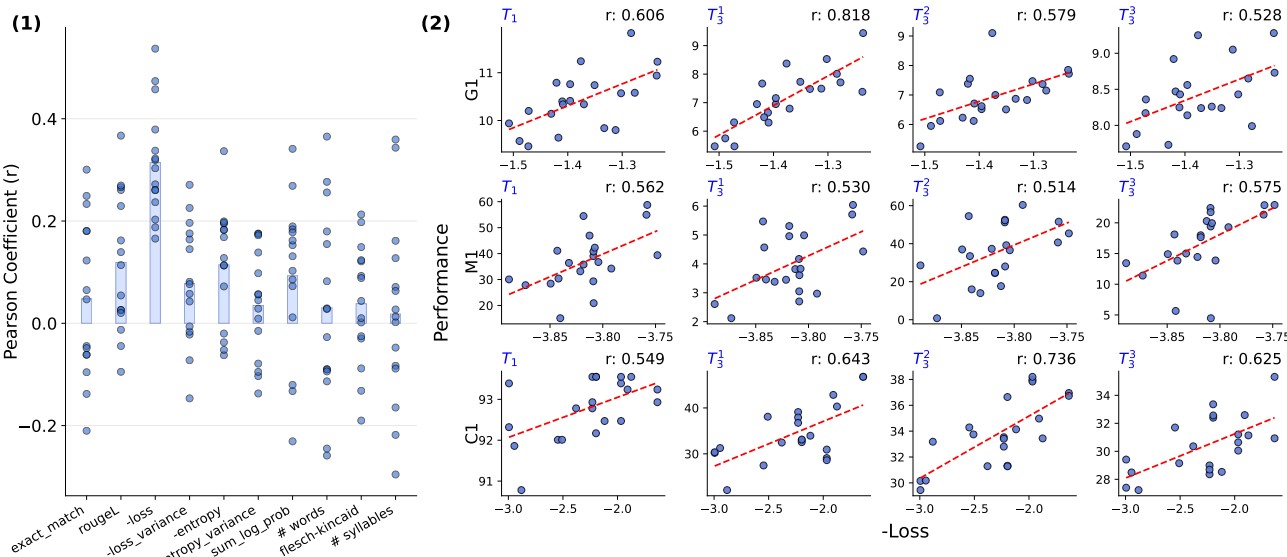

*Figure 3.* (1) Pearson correlations between 10 pre-update metrics and post-training cross-task performance. Each dot averages correlation across non-training tasks for a training pair, yielding 15 points per metric. Bar height denotes the mean of these 15 points. (2) Expanded view of the measurement with the highest average correlation: negative loss. Each row represents one task sequence and each subplot a downstream evaluation task. Each subplot shows negative pre-learning loss vs. post-learning performance across 21 training prompts.

optimization objective.

Our findings advance the operational understanding of fine-tuning by elucidating the pivotal role of training prompts. Recent studies posit that fine-tuning primarily modulates input-to-capability activation pathways rather than creating new capabilities, attributing cross-task performance drift to conflicts between pathways (Kotha et al., 2024; Zheng et al., 2025; Jiang et al., 2025). However, prior work overlooks the critical role of training prompts in mitigating such conflicts. Since distinct prompts occupy unique positions in the representation space, they establish activation paths that intersect differently with the functional manifolds of other tasks, leading to significant variance in their impact on forgetting and generalization. Crucially, these impacts are positively correlated, indicating that the training prompt can serve as an effective control mechanism to regulate the pathway conflicts. This highlights that training prompt engineering is not merely a surface-level adjustment, but a pivotal lever for orchestrating the model's global capability landscape during fine-tuning.

# 4. Methodology: State-Adaptive Prompt Optimization

Our systematic investigation has demonstrated the profound cross-task impact of training prompts and the existence of superior formulation, underscoring both the necessity and feasibility of training prompt engineering. Based on these foundations, we introduce State-Adaptive Prompt Optimization (SAPO), a lightweight training strategy designed to dynamically align prompt formulation with model's evolving state. SAPO utilizes a simple yet robust metric, task loss, to efficiently filter superior prompts prior to learning.

## 4.1. Identifying Superior Prompts via Pre-Update Loss

The cornerstone of an effective prompt engineering method is the ability to efficiently identify superior prompts before training. We frame this as a selection problem: identifying the optimal prompt from a candidate pool using quantitative indicators. To investigate whether there exist specific signals correlate strongly with post-training cross-task performance, a comprehensive search is conducted across three categories of potential signals: (1) **Prompt-intrinsic signals** focus solely on the text properties of the prompt, independent of the model state. Metrics include word count, syllable count, and readability scores such as Flesch–Kincaid grade (Kincaid et al., 1975); (2) **Model-behavior signals** reflect model's initial response using the prompt. We evaluate the pre-update loss (causal language modeling loss over the target outputs (Radford et al., 2019)), the total probability assigned to the outputs, and zero-shot performance metrics (e.g., Exact Match, Rouge-L (Lin, 2004)). They quantify the alignment between the specific prompt formulation and the model's knowledge; (3) **Uncertainty signals** capture model's instability when solving the task with the prompt, measured by the variance of the above quantities across training instances. These metrics are computed for all candidate prompts across the 120 task sequences described in § 3.3. Then we calculate the Pearson correlation between each pre-update metric and the model's post-training per-

**Algorithm 1** State-Adaptive Prompt Optimization (SAPO)

**Input:** Language model $\theta_0$, datasets $\{\mathcal{D}_t\}_{t=1}^N$, original prompts $\{P_t^0\}_{t=1}^N$, paraphraser $\mathcal{G}$
**Output:** Optimized language model $\theta_N$
1: **for** $t = 1, \ldots, N$ **do**
2:      # Prompt Expansion
3:      Generate prompt pool: $\mathcal{C}_t = \{P_t^{(k)}\}_{k=1}^K \leftarrow \mathcal{G}(P_t^0)$
4:      # State-Adaptive Alignment Evaluation
5:      Sample evaluation subset $\widetilde{\mathcal{D}}_t \subset \mathcal{D}_t$
6:      **for** each candidate prompt $P \in \mathcal{C}_t$ **do**
7:          Compute loss: $L(\theta_{t-1}, P)$
8:      **end for**
9:      # Optimized Formulation Integration
10:      Select prompt: $P_t^* \leftarrow \arg\min_{P \in \mathcal{C}_t} L(\theta_{t-1}, P)$
11:      Fine-tune model: $\theta_t \leftarrow \text{Train}(\theta_{t-1}, \mathcal{D}_t, P_t^*)$
12: **end for**
13: **return** $\theta_N$

formance on each of the nine non-training tasks ($T_1$ and $\{T_3^j\}_{j=1}^8$). The left panel of Figure 3 reports these correlations for Llama2-7b-chat model. For clarity, we average the correlations across these nine evaluation tasks for each training pair ($T_1, T_2$), yielding 15 representative data points per metric. The bar height represents mean of the 15 points.

Among all evaluated signals, the negative task loss exhibits the strongest positive association with post-learning cross-task performance. While other metrics, such as Rouge-L, show moderate correlation, the loss metric provides the most consistent and robust signal. Crucially, this correlation is uniformly non-negative, suggesting that selecting low-loss prompts is a safe strategy that generally improves, and at worst maintains, cross-task performance. Consequently, pre-update loss is a reliable proxy for identifying superior prompts. This observation is robust across diverse task categories (classification, generation, mixed), varying model families/sizes (Llama-2-7b-chat, Qwen3-8b, Qwen3-14b), and alternative correlation metrics (e.g., Spearman (Reimers et al., 2016)) . Complete analyses are in Appendix B.3 (see Figure 11–14). In summary, the choice of training prompt matters: not only does it significantly impact capabilities and allow for optimization, but the superior forms are efficiently identifiable prior to training.

### 4.2. State-Adaptive Prompt Optimization Method

Motivated by the insight that training prompts are impactful, optimizable, and the optimal formulation is identifiable via pre-update loss, we propose State-Adaptive Prompt Optimization (SAPO). SAPO is a lightweight, plug-and-play training strategy designed to dynamically align task instructions with the model's evolving state to mitigate forgetting and improve generalization. Specifically, prior to learning

a new task, SAPO executes the following steps, as detailed in Algorithm 1: **1. Prompt Expansion.** Leveraging a paraphrasing model (e.g., Gemini-2.5-Pro), a small pool of semantically equivalent prompts are generated based on the original task instruction. Our analysis in Appendix D indicates that a pool size of 20 is sufficient to capture effective prompt variations. **2. State-Adaptive Alignment Evaluation.** The pre-update loss for each candidate prompt is computed using current task's training subset. This score serves as a proxy for the alignment between prompt formulation and model's current state. Notably, this step requires only a forward pass on a subset with a small pool of candidates, which adds limited overhead relative to training and ensures efficiency. **3. Optimized Formulation Integration.** The prompt with the lowest loss is selected for the subsequent fine-tuning phase. Crucially, this same prompt is consistently used for the evaluation of the task.

The distinct characteristic of SAPO lies in its shift from static data consumption to dynamic, state-adaptive task formulation Unlike traditional paradigms that treat training data as fixed artifacts, SAPO views the task instantiation as an optimizable variable dependent on the model's state. By prioritizing input prompts with lower pre-update loss, SAPO ensures that the training context remains aligned with model's intrinsic distribution and current knowledge base. As detailed in our mechanism analysis (§ 5.5), this alignment makes better use of model's existing capabilities and minimizes conflicting task-specific adaptations, effectively mitigating forgetting and enhancing generalization. Therefore, SAPO is orthogonal to existing fine-tuning algorithms, and can be seamlessly integrated to transform fixed, state-agnostic training processes into state-adaptive ones.

## 5. Experiments

We conduct comprehensive empirical experiments on the continual learning setting to demonstrate the effectiveness of our SAPO method, which corroborates our findings regarding the critical role of training prompts. Through further ablation study and analysis, we clarify that the efficacy of SAPO stems from the selection of low-loss prompts, which guides the model to acquire more generalizable knowledge.

### 5.1. Experimental Settings

**Benchmarks**. Following our probing setup, we construct continual instruction-tuning sequences using SuperNI (Wang et al., 2022), each with 5 tasks. We instantiate three sequence types: homogeneous classification, homogeneous generation, and mixed (alternating classification and generation), with two sequences per type. To assess robustness beyond these controlled settings, we extend our evaluation to the benchmark TRACE (Wang et al., 2023b), which incorporates a more heterogeneous sequence of six learning

*Table 1.* Performance of continual learning methods and their state-adaptive version with SAPO on four benchmarks.

| | Method | NI-Seq-G1 | | | NI-Seq-C1 | | | NI-Seq-M1 | | | TRACE | | |
|---|---|---|---|---|---|---|---|---|---|---|---|---|---|
| | | AP ↑ | BWT ↑ | FWT ↑ | AP ↑ | BWT ↑ | FWT ↑ | AP ↑ | BWT ↑ | FWT ↑ | AP ↑ | BWT ↑ | FWT ↑ |
| **Llama2-7b-chat** | LoraInc | 35.88 | -5.63 | 6.87 | 76.54 | -0.57 | 4.39 | 59.13 | -3.86 | 7.31 | 49.13 | -9.22 | 19.3 |
| | +SAPO | +2.36 | +0.56 | -0.13 | +0.16 | +0.49 | +1.76 | +2.92 | +1.70 | +5.62 | +0.63 | +3.27 | -0.34 |
| | EWC | 35.58 | -5.40 | 6.22 | 73.30 | -0.74 | 6.63 | 60.27 | -1.21 | 10.92 | 46.12 | -4.92 | 13.54 |
| | +SAPO | +4.93 | +1.4 | +0.21 | +1.43 | +0.23 | +1.39 | +1.20 | +0.36 | +3.37 | +2.87 | +0.34 | +1.65 |
| | O-Lora | 43.45 | -2.39 | 8.92 | 71.27 | -0.56 | 4.05 | 60.29 | -0.61 | 6.53 | 45.88 | -7.04 | 20.1 |
| | +SAPO | +1.30 | +0.31 | +0.51 | +1.23 | +0.75 | +0.30 | +1.12 | -0.26 | +0.49 | +2.45 | +2.56 | +0.98 |
| | InsCL | 45.53 | -2.29 | 7.87 | 75.89 | -0.25 | 3.31 | 62.29 | -1.68 | 10.07 | 51.62 | -3.32 | 15.70 |
| | +SAPO | +0.29 | -0.17 | +0.22 | +0.69 | +0.3 | +0.02 | +1.71 | +0.29 | +1.93 | +0.76 | +0.59 | +0.92 |
| **Qwen3-8b** | LoraInc | 45.08 | -1.33 | -2.14 | 80.16 | 0.34 | -8.00 | 65.41 | -2.06 | 4.16 | 58.21 | -5.86 | 14.84 |
| | +SAPO | +0.58 | +0.44 | -0.23 | +0.62 | +0.18 | +1.98 | +1.89 | +0.5 | +0.06 | +0.72 | +1.03 | +1.62 |
| | EWC | 44.14 | -1.30 | -2.45 | 78.54 | -0.44 | -5.93 | 62.17 | -0.39 | 0.23 | 55.68 | -4.78 | 10.38 |
| | +SAPO | +1.23 | +0.82 | +1.97 | +1.73 | +0.58 | +1.88 | +2.45 | -0.26 | +2.55 | +1.66 | +1.29 | +1.64 |
| | O-Lora | 43.05 | -1.81 | -1.67 | 76.95 | 0.42 | -6.99 | 65.05 | -1.61 | -1.40 | 54.90 | -7.48 | 6.65 |
| | +SAPO | +1.72 | +1.38 | +0.30 | +0.57 | +0.16 | +1.64 | +1.07 | +0.53 | +2.97 | +2.72 | +3.43 | +2.76 |
| | InsCL | 46.10 | -1.22 | -2.23 | 79.40 | -0.43 | -11.00 | 66.04 | -1.35 | 5.39 | 58.83 | -4.24 | 12.89 |
| | +SAPO | +0.33 | +0.98 | +1.23 | +0.54 | +0.43 | +0.30 | +0.95 | +0.29 | +0.16 | +0.79 | +1.05 | +0.92 |
| **Llama2-13b** | LoraInc | 39.33 | -5.12 | 11.18 | 76.91 | -1.09 | 23.32 | 64.17 | -15.46 | 24.96 | 51.78 | -11.95 | 12.94 |
| | +SAPO | +1.48 | +0.76 | +0.28 | +0.29 | +0.12 | +2.69 | +2.17 | +0.55 | +2.27 | +1.17 | +1.62 | +1.35 |
| | O-Lora | 44.52 | -2.09 | 7.96 | 76.52 | -1.51 | 22.53 | 63.68 | -1.55 | 29.08 | 46.58 | -9.09 | 10.73 |
| | +SAPO | +0.76 | +1.11 | +1.32 | +0.94 | -0.12 | +1.35 | +1.45 | +0.61 | +0.37 | +2.85 | +1.13 | +1.76 |
| **Qwen3-14b** | LoraInc | 44.04 | -0.75 | -1.42 | 80.00 | -0.32 | 10.45 | 65.50 | -4.07 | 9.72 | 57.93 | -6.33 | -7.74 |
| | +SAPO | +1.95 | +0.56 | +1.20 | +0.43 | +0.48 | +1.72 | +1.65 | +0.73 | -0.36 | +0.74 | +1.23 | +1.58 |
| | O-Lora | 46.42 | -0.27 | -1.07 | 78.61 | -0.20 | 12.66 | 66.89 | -1.9 | 9.95 | 55.64 | -7.47 | -5.91 |
| | +SAPO | +0.94 | +0.35 | +1.45 | +1.02 | +0.12 | +1.76 | +0.43 | +0.63 | +1.89 | +2.15 | +2.33 | +0.92 |

tasks. Notably, it features complex tasks such as mathematical reasoning and code generation, which heavily rely on the core capabilities of modern LLMs. Full construction details appear in Appendix C.

**Evaluation Metrics**. Rouge-L (Lin, 2004) is utilized as the unified performance metric for both classification and generation tasks. For classification, Rouge-L aligns with standard accuracy via output processing (Zhao et al., 2024a). Three widely used metrics are adopted to quantify different aspects of continual learning dynamics (Chaudhry et al., 2018; Buzzega et al., 2020; Pan et al., 2025). For a sequence of $N$ tasks, let $a_{i,j}$ denote the test performance on task $j$ after the model has finished training on task $i$, the metrics are defined as: **(1) AP** $= \frac{1}{N}\sum_{j=1}^{N} a_{N,j}$. Average Performance averages the model's final performance over all tasks after completing the training sequence, reflecting overall ability acquisition and retention. **(2) BWT** $= \frac{1}{N}\sum_{i=2}^{N}\frac{1}{i-1}\sum_{j=1}^{i-1} a_{i,j} - a_{i-1,j}$. Backward Transfer averages the step-wise change in performance on previously learned tasks. It quantifies how learning the i-th task impacts knowledge retained from prior tasks. Since it typically takes negative values, it indicates forgetting on trained tasks. **(3) FWT** $= \frac{1}{N}\sum_{i=1}^{N-1}\frac{1}{N-i}\sum_{j=i+1}^{N} a_{i,j} - a_{i-1,j}$. Forward Transfer averages the step-wise change in performance on future (unseen) tasks. It quantifies how learning the i-th task influences capabilities required for subsequent tasks, serving as a proxy for generalization.

**Comparison methods**. Our approach is evaluated against representative state-of-the-art continual learning methods spanning three primary families. **(1) Model modularization**: **LoraInc** (Hu et al., 2022) incrementally adds and updates new task-specific LoRA parameters; **O-LoRA** (Wang et al., 2023a) extends LoraInc by constraining updates for new LoRA parameters to be orthogonal to previous learned ones. **(2) Parameter regularization**: **EWC** (Elastic Weight Consolidation) (Huang et al., 2024) uses Fisher information to estimate parameter importance and penalize shifts in important parameters. **(3) Data replay**: **InsCL** (Wang et al., 2024) maintains exemplars from prior tasks and employs LLM-based filtering to optimize for quality and diversity.

**Model fine-tuning**. We conduct continual fine-tuning experiments across four distinct language models: Llama-2-7b-chat, Llama-2-13b-chat (Touvron et al., 2023), Qwen3-8b, and Qwen3-14b (Yang et al., 2025), using the standard causal language modeling loss (Radford et al., 2019). Unless otherwise specified, we employ LoRA fine-tuning (Hu et al., 2022), with the Adam optimizer, epoch 10, learning rate 1e-4, and batch size 64. Additional implementation details can be found in Appendix C.1.

### 5.2. Main Results

Table 1 presents the continual learning performance on four benchmarks, leading to several key observations. **1) SAPO**

*Table 2.* Comparison of performance effects between state-adaptive prompt optimization (SAPO) and pessimization (SAPP).

| | Method | NI-Seq-G1 | | | NI-Seq-C1 | | |
|---|---|---|---|---|---|---|---|
| | | AP | BWT | FWT | AP | BWT | FWT |
| **Llama2-7b-chat** | LoraInc | 35.88 | -5.63 | 6.87 | 76.54 | -0.57 | 4.39 |
| | +SAPP | -0.67 | -1.05 | -0.25 | -1.56 | -0.47 | -0.88 |
| | +SAPO | +2.36 | +0.56 | -0.13 | +0.16 | +0.49 | +1.76 |
| | O-Lora | 43.45 | -2.39 | 8.92 | 71.27 | -0.56 | 4.05 |
| | +SAPP | -1.24 | -1.17 | -0.22 | -0.75 | -1.12 | -1.08 |
| | +SAPO | +1.30 | +0.31 | +0.51 | +1.23 | +0.75 | +0.30 |
| **Qwen3-8b** | LoraInc | 45.08 | -1.33 | -2.14 | 80.16 | 0.34 | -8.00 |
| | +SAPP | -1.14 | -0.98 | -0.39 | -1.38 | -0.81 | -1.57 |
| | +SAPO | +0.58 | +0.44 | -0.23 | +0.62 | +0.18 | +1.98 |
| | O-Lora | 43.05 | -1.81 | -1.67 | 76.95 | 0.42 | -6.99 |
| | +SAPP | -1.27 | -0.64 | +0.21 | -0.67 | -1.33 | -2.56 |
| | +SAPO | +1.72 | +1.38 | +0.30 | +0.57 | +0.16 | +1.64 |

*Table 3.* Performance improvements of SAPO using different paraphraser models. Results are averaged over 3 runs on Qwen3-8b trained on the NI-Seq-M1 sequence.

| Method | Para. | $\Delta$ AP $\uparrow$ | $\Delta$ BWT $\uparrow$ | $\Delta$ FWT $\uparrow$ |
|---|---|---|---|---|
| **LoraInc** | Gemini | +1.89 | +0.50 | +0.06 |
| | GPT | +1.47 | +1.21 | +0.31 |
| | Qwen | +1.66 | +0.62 | +0.05 |
| **O-Lora** | Gemini | +1.07 | +0.53 | +2.97 |
| | GPT | +1.11 | +0.47 | +1.79 |
| | Qwen | +0.55 | +0.76 | +2.49 |

**universally improves the performance of all continual learning (CL) methods**. While traditional CL strategies all aim to mitigate catastrophic forgetting, their effectiveness varies significantly in LLM setting. Considering backward transfer (BWT), InsCL generally achieves the best performance, with other methods lagging behind. Similar trends are also observed in the average performance (AP). Despite these wide performance disparities, SAPO yields uniform improvements across all metrics for every method. SAPO brings significant gains to weak baselines while advancing the performance of the strongest. These universal gains validate our core insight: adapting the task formulation to the model's current state is a critical yet previously overlooked factor in optimizing LLM learning dynamics.

**2) SAPO is robust across diverse task sequences**. Continual learning performance varies significantly across task sequences, driven by inter-task similarity. Low-similarity sequences, such as Trace benchmark and NI-Seq-G1/M1 sequences, suffer from more severe forgetting (lower BWT). In contrast, high-similarity sequences, such as NI-Seq-C1 (where all tasks involve selection from options), exhibit better retention and generalization. Crucially, SAPO consistently improves performance across all these scenarios. This demonstrates the broad applicability and robustness of our state-adaptive mechanism: by optimizing for the model's immediate state, SAPO remains effective regardless of the high-level semantic properties of the task sequence.

### 5.3. Ablation Study: Efficacy of Prompt Optimization

Our main results in Table 1 show that SAPO significantly outperforms training with fixed human-authored prompts. To verify that these gains stem from the active optimization process rather than simply avoiding potentially poor-quality human prompts, an ablation is conducted using a state-adaptive prompt pessimization (SAPP) strategy. In SAPP, we generate paraphrased candidates but deliberately select the prompt yielding the highest pre-learning task loss. As shown in Table 2, this adversarial selection leads to a

consistent performance decline across all models and sequences. This confirms that the improvements of our SAPO strategy are not simply from avoiding sub-optimal human prompts; rather, they are driven by the specific efficacy of the state-adaptive mechanism. Furthermore, it validates the pre-learning loss as a reliable signal for prompt quality.

### 5.4. Ablation Study: Robustness to Paraphrasers

SAPO utilizes paraphraser models to generate semantically equivalent candidate prompts. Since producing such simple semantic variations is a trivial task that modern LLMs can easily accomplish, SAPO does not rely on a specific or exceptionally powerful paraphraser to be effective. To empirically validate this, we conduct an ablation study using three distinct LLMs of varying capabilities as paraphrasers: Gemini-2.5-Pro, GPT-OSS-120B, and Qwen3-32B (Comanici et al., 2025; OpenAI, 2025; Yang et al., 2025). Table 3 reports the results, averaged over 3 runs on the Qwen3-8b model using the NI-Seq-M1 sequence. As shown, SAPO yields consistent improvements in average performance (AP), backward transfer (BWT), and forward transfer (FWT) across all three paraphrasers when applied to both LoraInc and O-Lora baselines. These sustained gains confirm that SAPO's efficacy stems fundamentally from its state-adaptive selection mechanism, proving that the framework remains highly robust without demanding advanced prompt generation capabilities.

### 5.5. Mechanism Analysis: Adaptive Alignment Mitigates Optimization Conflicts

To elucidate why SAPO's adaptive alignment, achieved via lower-loss training prompts, mitigates forgetting and enhances generalization, we analyze its impact on model's intrinsic learning dynamics. We employ the inter-task gradient angles to quantify the degree of conflict and synergy in the learning process (Yu et al., 2020; Fifty et al., 2021). Specifically, for the trained Llama2-7b-chat model ($M_1$), we compute cosine similarities between the gradients of current task $T_2$ (conditioned on varying prompts) and those of non-training tasks $T_1$ and $T_3$. Figure 4 illustrates the evolution of these similarity distributions as loss decreases,

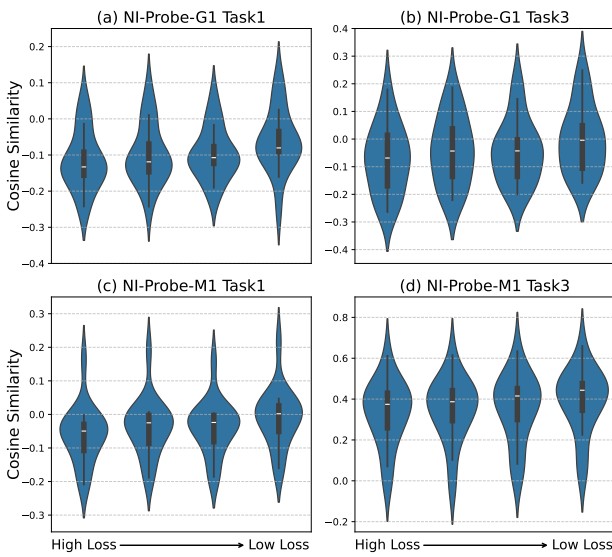

*Figure 4.* Changes in gradient cosine similarity distributions between Task 2 and Task 1/3 as Task 2 prompt loss decreases.

using four representative prompts to span the full loss spectrum of the 20 paraphrased candidates. The violin plots depict the distribution of gradient angles across different model modules (see Appendix C.5 for full details). A clear trend is observed: as prompt loss decreases, the angle between the gradients shrinks (i.e., their similarity increases). This geometric alignment indicates that low-loss training prompts effectively minimize the optimization conflict between tasks, thereby potentially facilitating the acquisition of more generalizable knowledge.

These geometric findings align with intuitive expectations. A model's capabilities can be conceptualized as a combination of general and task-specific abilities, both of which are updated during learning (Huang et al., 2021). In this context, a high-loss training prompt typically indicates vague task instruction with sparse effective information, forcing the model to internalize substantial task-specific knowledge to bridge the gap, which increases the likelihood of inter-task conflicts. In contrast, a low-loss prompt implies that the necessary task-specific logic is largely encoded within the instruction. By better leveraging the model's intrinsic capabilities, these prompts relieve the gradient updates of learning extensive specific adaptations, effectively mitigating potential conflicts. Consequently, SAPO enforces this adaptive alignment to steer the learning trajectory toward maximal compatibility with the model's broader capability landscape, reducing interference and enhancing transfer.

## 6. Conclusion

In this work, we identify training prompt formulation as a critical yet underexplored dimension in LLM fine-tuning.

Our analysis reveals a deceptive consistency: while semantically equivalent prompts yield comparable in-task performance, they induce drastically different outcomes regarding forgetting and generalization. Crucially, this variability is systematic and predictable, allowing for the identification of superior prompts via the pre-update loss. Building on these insights, we propose State-Adaptive Prompt Optimization (SAPO), a lightweight strategy that dynamically aligns task instructions with model's evolving state. Mechanistically, this state alignment reduces inter-task gradient conflicts, potentially facilitating acquisition of generalizable knowledge. We believe our work highlights the importance of state-aware data formulation, opening new avenues that extend beyond robust LLM fine-tuning to wider training scenarios.

## Acknowledgements

The work is supported in part by the National Natural Science Foundation of China (NSFC) under Grant 62441230, 62502522, 62072458, 62472429 and 62461146205, and in part by the Outstanding Innovative Talents Cultivation Funded Programs 2024 of Renmin University of China.

## Impact Statement

Fine-tuning serves as the predominant paradigm for adapting LLMs to downstream domains and tasks. In this work, we identify training prompt formulation as a critical factor in model stability and introduce SAPO to dynamically optimize these interactions. By mitigating catastrophic forgetting and enhancing generalization, SAPO ensures that models adapt to specific tasks without degrading their core general competencies. Crucially, this stability extends to safety alignment, serving as a structural safeguard against alignment drift by helping to preserve pre-existing safety guardrails and ethical constraints. Collectively, these improvements foster the development of more versatile and trustworthy AI systems suitable for widespread real-world deployment.

Beyond its positive impacts, SAPO could potentially have negative consequences. While SAPO enhances task learning dynamics during fine-tuning, this capability is content-agnostic and could theoretically be employed to improve training efficiency on malicious datasets. Therefore, as with all advancements in LLM training methodologies, responsible data curation and robust monitoring remain essential prerequisites for deployment.

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

# A. Probe Datasets

Our investigation is conducted on datasets derived from the SuperNI benchmark (Wang et al., 2022), which is widely utilized in existing instruction-following works. We select 26 tasks from the original benchmark. For each task, we set both the training and testing set sizes to 1,000 samples. The statistical information for these tasks is listed in Table 5. Based on these tasks, we construct 120 three-task sequences $(T_1, T_2, T_3)$, corresponding to the previously trained, current target, and unseen tasks, respectively. We create 40 sequences for each of three sequence types: generation-only (G), classification-only (C), and mixed (M). These 40 sequences for each type are generated by combining 5 distinct pairs of trained $(T_1)$ and current $(T_2)$ tasks with 8 distinct unseen $(T_3)$ tasks $(5 \times 8 = 40)$. The composition of each three-task sequence is enumerated in Table 6. In Figure 1, we report the performance on one generalization task from each of the G1, M1, and C1 sequences. These tasks—task1355, task224, and task1343, respectively—are bolded in Table 6 for reference.

*Table 4.* A total of 120 three-task sequences are used for the probe experiments. These consist of three sequence types, with 40 sequences each: pure generation, pure classification, and a mixture of generation and classification. In the mixture sequences, classification and generation tasks appear alternately.

|  | Sequence | Task Type |
|---|---|---|
| NI-Probe-G1 | NI589 → NI339 → NI{618, 511, 1290, **1355**, 163, 488, 24, 141 } | Generation |
| NI-Probe-G2 | NI589 → NI618 → NI{163, 339, 292, 141, 2, 511, 24, 488} | Generation |
| NI-Probe-G3 | NI141 → NI589 → NI{163, 360, 488, 511, 339, 618, 24} | Generation |
| NI-Probe-G4 | NI141 → NI618 → NI{163, 339, 292, 24, 511, 2, 589, 488} | Generation |
| NI-Probe-G5 | NI339 → NI618 → NI{163, 292, 141, 511, 24, 619} | Generation |
| NI-Probe-C1 | NI1310 → NI231 → NI{1510, 220, 611, **224**, 363, 1292, 195, 273} | Classification |
| NI-Probe-C2 | NI1292 → NI224 → NI{1510, 611, 273, 220, 231, 363, 195, 1310} | Classification |
| NI-Probe-C3 | NI1292 → NI1310 → NI{1510, 611, 224, 231, 273, 220, 195, 363} | Classification |
| NI-Probe-C4 | NI273 → NI1292 → NI{1510, 231, 220, 611, 1310, 224, 195, 363} | Classification |
| NI-Probe-C5 | NI231 → NI1292 → NI{1510, 224, 611, 363, 1310, 273, 195, 220} | Classification |
| NI-Probe-M1 | NI1292 → NI618 → NI{163, 363, 224, **1343**, 195, 1310, 611, 339 } | Classification & Generation |
| NI-Probe-M2 | NI1292 → NI618 → NI{195, 363, 1310, 611, 224, 163, 339, 231} | Classification & Generation |
| NI-Probe-M3 | NI611 → NI618 → NI{163, 1292, 224, 363, 195, 231, 1310, 339} | Classification & Generation |
| NI-Probe-M4 | NI618 → NI611 → NI{163, 24, 360, 224, 339, 1292, 141, 488} | Generation & Classification |
| NI-Probe-M5 | NI589 → NI195 → NI{163, 1292, 1310, 224, 360, 611, 141, 363} | Generation & Classification |

# B. Supplementary Probe Experiments

In our main paper (§3, 3.3, and 4.1), we conduct three probe experiments to systematically investigate the necessity of training prompt engineering. In this section, we provide supplementary experimental results across a broader range of models and task sequences to further demonstrate the robustness and reliability of our findings and conclusions.

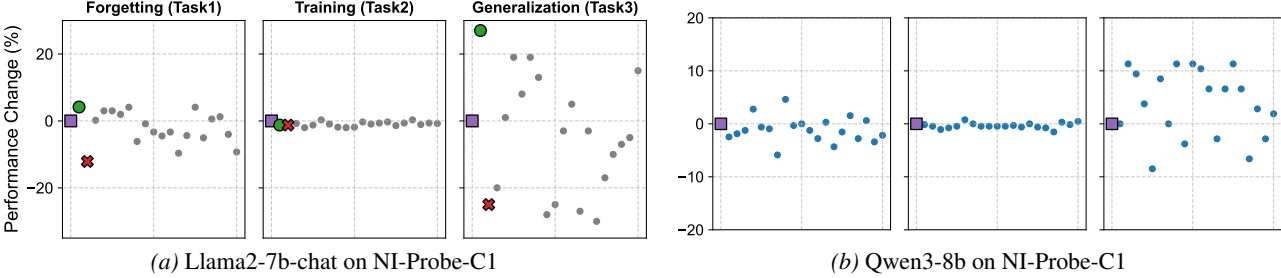

*(a)* Llama2-7b-chat on NI-Probe-C1    *(b)* Qwen3-8b on NI-Probe-C1

*Figure 5.* Normalized relative performance change (vs. the original prompt) on the trained, current, and unseen tasks after training with semantically equivalent paraphrased prompts. Results shown for a classification sequence on Llama-2-7b-chat and Qwen3-8b. ■ marks the original prompt. The three prompts marked for Llama-2-7b-chat are shown in Table 1.

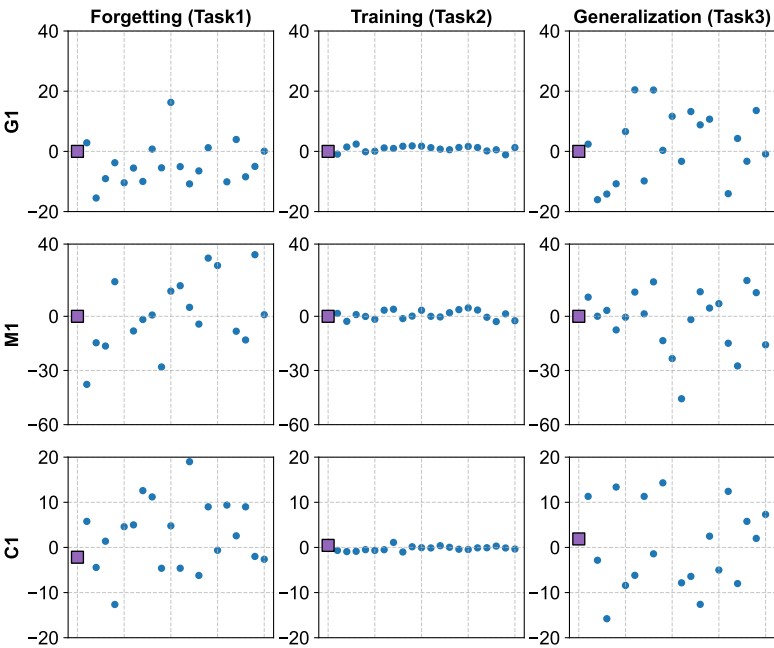

*Figure 6.* Normalized relative performance change (vs. the original prompt) on the trained, current, and unseen tasks after training with semantically equivalent paraphrased prompts. Results shown for three sequences on Qwen3-14b.

### B.1. Divergent Cross-Task Impacts

In Figure 1, we present the performance variations on Llama2-7b-chat and Qwen3-8b when using paraphrased prompts on a generative sequence and a mixed sequence. Furthermore, in Figure 5, we illustrate the performance variations for these two models on a classification-only sequence. Additionally, we present results for the larger Qwen3-14b model on the same generative, mixed, and classification sequences in Figure 6. Across all tested model families, model sizes, and task sequences, our central finding holds robustly: the choice of training prompts has a negligible impact on in-task performance but significantly affects catastrophic forgetting on previously trained tasks and generalization to unseen tasks.

### B.2. Existence of Superior Training Prompts

In Figure 2, we present the pairwise task performance correlations for Llama2-7b-chat and Qwen3-8b on the NI-Probe-G1 and NI-Probe-M1 sequences, as measured by the Pearson correlation coefficient. In Figure 7, we provide the corresponding correlation analyses for these models and sequences using the Spearman coefficient. Furthermore, in Figures 8 and 9, we illustrate the pairwise correlations measured by both Pearson and Spearman coefficients, respectively, for these two models on a classification-only sequence. Additionally, we present results for the larger Qwen3-14b model on the same generative, mixed, and classification sequences in Figure 10. Across all tested model families, model sizes, task sequences, and correlation metrics, the strong positive correlation between task performances holds robustly. This supports our conclusion that **better training prompts exist which consistently improve cross-task performance**.

### B.3. Identifying Superior Prompts via Pre-Update Loss

In Figure 3, we present the Pearson correlations between 10 pre-learning measurements and the post-learning performance for Llama2-7b-chat model. And in Figure 11, we provide the corresponding correlation analyses for Llama2-7b-chat model using the Spearman coefficient. Furthermore, in Figures 12 and 13, we present the corresponding correlations measured by both Pearson and Spearman coefficients for another model, Qwen3-8b. Additionally, we present results for the larger Qwen3-14b model on the same 120 sequences in Figure 14. Across all tested model families, model sizes, and correlation metrics, we consistently find that the pre-learning negative task loss exhibits a strong positive correlation with post-learning performance on non-training tasks. This robustly confirms our conclusion: the pre-task loss, computed on the training set, can be used to identify better-performing prompts before the learning process begins.

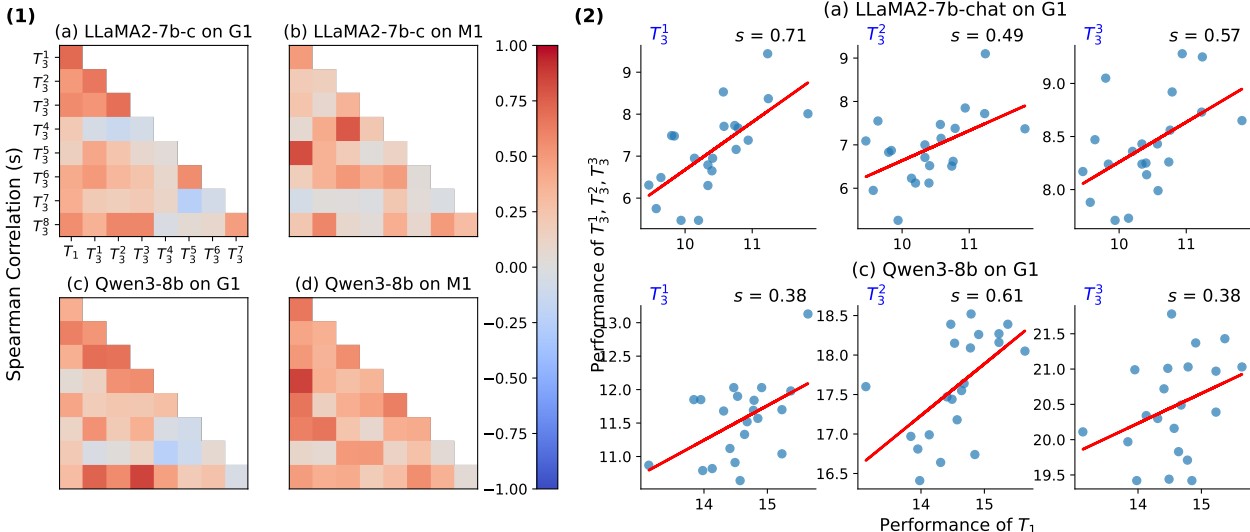

*Figure 7.* Pairwise Spearman correlations among performances across the trained task $T_1$ and eight unseen tasks $\{T_3^j\}_{j=1}^8$. Each subplot shows a combination between Llama2-7b-chat/Qwen3-8b model and a generative/mixed sequence

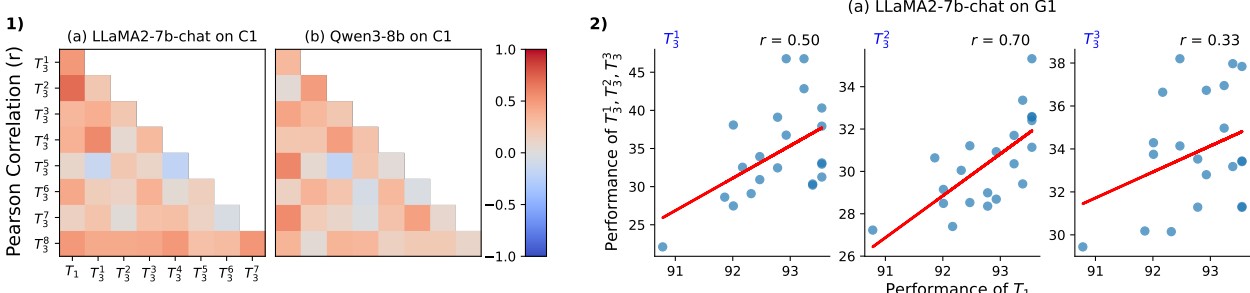

*Figure 8.* Pairwise Pearson correlations among performances across the trained task $T_1$ and eight unseen tasks $\{T_3^j\}_{j=1}^8$. Each subplot shows the results of Llama2-7b-chat and Qwen3-14b on a classification sequence.

## C. Details of Empirical Experiments

### C.1. Training and Evaluation

We adopt Llama2-7b-chat, Llama2-13b-chat (Touvron et al., 2023), Qwen3-8b, and Qwen3-14b (Yang et al., 2025) as our base models. These models are selected for their proven effectiveness in both world knowledge understanding and instruction following. For each task in the sequence, we train the models using the standard causal language model loss (Radford et al., 2019). We optimize the models using the Adam optimizer with a cosine learning rate schedule and a warm-up phase. All models are trained for 10 epochs with a learning rate of 1e-4. We use a per-GPU batch size of 4 and 2 gradient accumulation steps. Training is conducted on 8 H20 GPUs utilizing the Deepspeed Zero2 framework (Rajbhandari et al., 2020). The maximum input and output sequence lengths are set to 1536 and 128, respectively. We employ the LoRA fine-tuning methodology (Hu et al., 2022), setting the rank dimension to 8 and targeting the query and value weight matrices. For the LoraInc, O-LoRA, and InsCL baselines, a new adapter is initialized for each new task, while all previous LoRA adapters are frozen. In contrast, for EWC, a single, larger adapter (rank 40) is initialized and continually updated throughout the entire task sequence.

We evaluate the performance on all tasks using Rouge-L (Lin, 2004). Following (Zhao et al., 2024a), classification accuracy is measured via ROUGE-L with appropriate output post-processing. To ensure deterministic generation, we set the temperature to 0 for all evaluations. All reported results for Llama2-7b-chat and Qwen3-8b are the average of two experimental runs with different random seeds. Experiments on the larger Llama2-13b-chat and Qwen3-14b models are

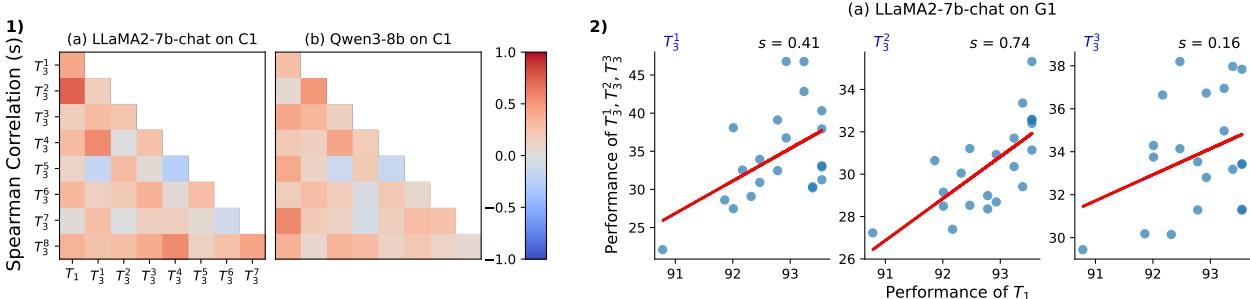

*Figure 9.* Pairwise Spearman correlations among performances across the trained task $T_1$ and eight unseen tasks $\{T_3^j\}_{j=1}^8$. Each subplot shows the results of Llama2-7b-chat and Qwen3-14b on a classification sequence.

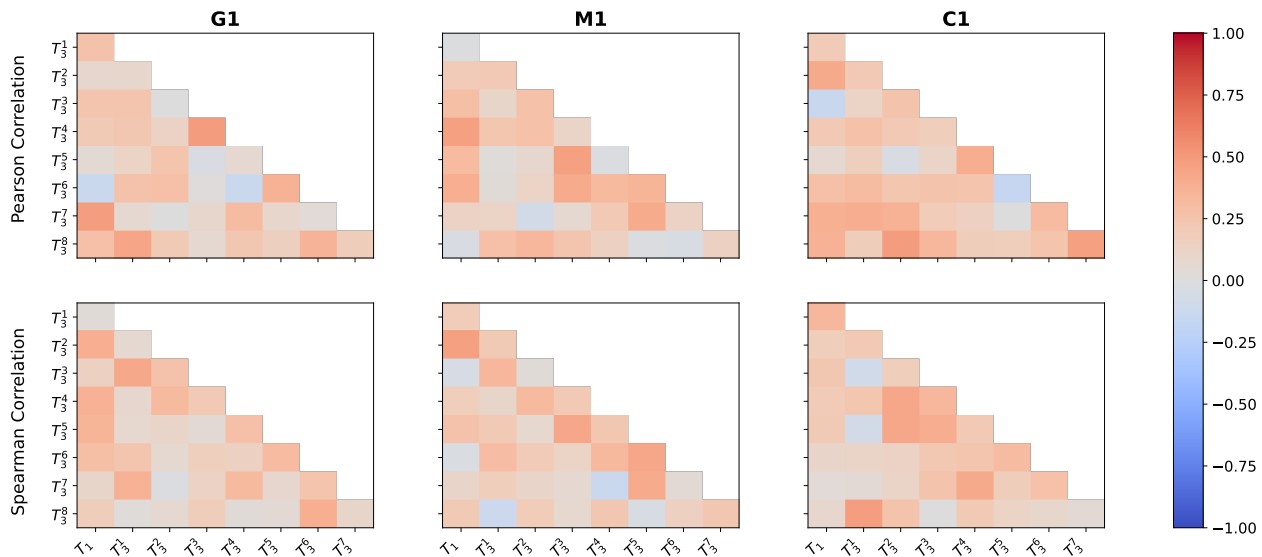

*Figure 10.* Pairwise Pearson and Spearman correlations among performances across the trained task $T_1$ and eight unseen tasks $\{T_3^j\}_{j=1}^8$. Each subplot shows the results of Qwen3-14b on a single sequence.

conducted with a single run, where we observed no anomalous results during these runs.

For the Llama2-7b-chat and Qwen3-8b models, we report comparisons against the full suite of baseline methods. For the larger 13B and 14B scales, due to computational constraints, we compare against model modularization methods (LoraInc and O-LoRA). We select this category as the representative baseline for two key reasons. First, these methods are grounded in Parameter-Efficient Fine-Tuning (PEFT) paradigm (Hu et al., 2022; Xu et al., 2023), which has emerged as the dominant paradigm for adapting large-scale models. Second, unlike traditional approaches that rely on historical data, methods like LoraInc offer broad applicability beyond strict continual learning constraints, supporting direct fine-tuning scenarios without such dependencies.

## C.2. SuperNI Benchmark

Consistent with our probe experiments, we conduct our primary empirical experiments on task sequences constructed from the SuperNI benchmark (Wang et al., 2022). For each of the three main task categories, we construct two distinct 5-task sequences. Detailed information about these sequences is listed in Table 9.

## C.3. Trace Benchmark

The TRACE benchmark (Wang et al., 2023b) is introduced for studying continual learning in LLMs. It comprises 8 diverse tasks, including multi-choice QA, code generation, mathematical reasoning, and summarization. Furthermore, the

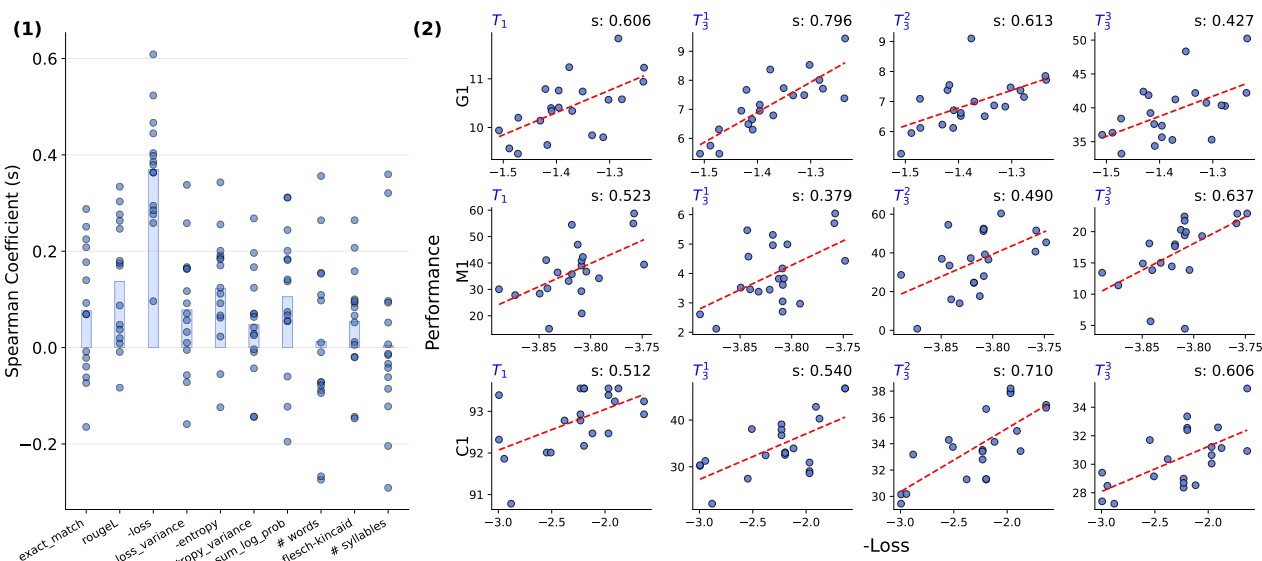

*Figure 11.* Spearman correlations between 10 pre-learning measurements and post-learning performance on other tasks. Results for Llama-2-7b-chat over 120 task sequences.

benchmark is multilingual, covering tasks in English, Chinese, and German. Following previous work (Jiang et al., 2025), we select 6 of the 8 tasks to construct our training sequence. Statistical details of the selected datasets are provided in Table 8. Unlike the SuperNI benchmark, where 1,000 samples are used per task, we utilize 3,000 samples per task for training on TRACE. Performance on all tasks is similarly evaluated using the ROUGE-L metric.

## C.4. Implementation Details

We compare our method against representative state-of-the-art (SOTA) continual learning methods from the three primary families. For each baseline, we perform a grid search to determine the optimal hyperparameters. **LoraInc** (Hu et al., 2022) incrementally adds and trains new task-specific LoRA parameters. This method requires no additional hyperparameters. **O-LoRA** (Wang et al., 2023a) builds on LoraInc, constraining updates for new LoRA parameters to be orthogonal to previously learned ones. The coefficient for its regularization term is set to 0.5. In, **EWC** (Elastic Weight Consolidation) (Huang et al., 2024), we set the scaling factor for the regularization term to 4,000. In **InsCL** (Wang et al., 2024), we maintain a fixed-size total replay buffer of M=200 exemplars, and employ the InsInfo metric, implemented via Gemini-2.5-Pro (Wang et al., 2024) scoring, to select the most representative samples.

Our SAPO method sets the candidate pool size to 20. For the **Prompt Expansion** step, we utilize Gemini-2.5-Pro (Comanici et al., 2025) to paraphrase the original task instruction, using the meta-prompt detailed in Table 7. For the **State-Adaptive Alignment Evaluation**, we assess candidate prompts using a subset of the training data to ensure efficiency. Specifically, for the SuperNI dataset (containing 1,000 samples per task), we evaluate on a randomly sampled subset of 250 instances. Similarly, for the TRACE dataset (3,000 samples per task), we utilize a subset of 1,000 instances. Taking SuperNI as an example, our fine-tuning involves forward and backward passes over 1,000 samples for 10 epochs in a training task. In contrast, SAPO requires only forward passes on $250 \times 20$ instances per task. Considering that the forward process requires no gradient computation or storage, allowing for significantly larger batch sizes compared to training, the additional time overhead introduced by SAPO is minor relative to the total training budget.

## C.5. Experimental Details of Low-Loss Prompts' Mechanism Analysis

In § 5.5, we present the analysis of gradient angles between the target task $T_2$ (using various prompts) and other tasks ($T_1$, $T_3$), demonstrating how this angle varies with prompt loss. Specifically, we follow the setup in § 4.1, using the Llama2-7b-chat model trained on $T_1$ ($M_1$) and two different task sequences. We compute two categories of gradients: (1) Target Task ($T_2$): Gradients are derived from four representative prompts selected to span the full loss ranking spectrum (specifically, the candidates ranked 1st, 7th, 13th, and 20th out of the 20 paraphrased options). (2) Other Tasks ($T_1, T_3$):

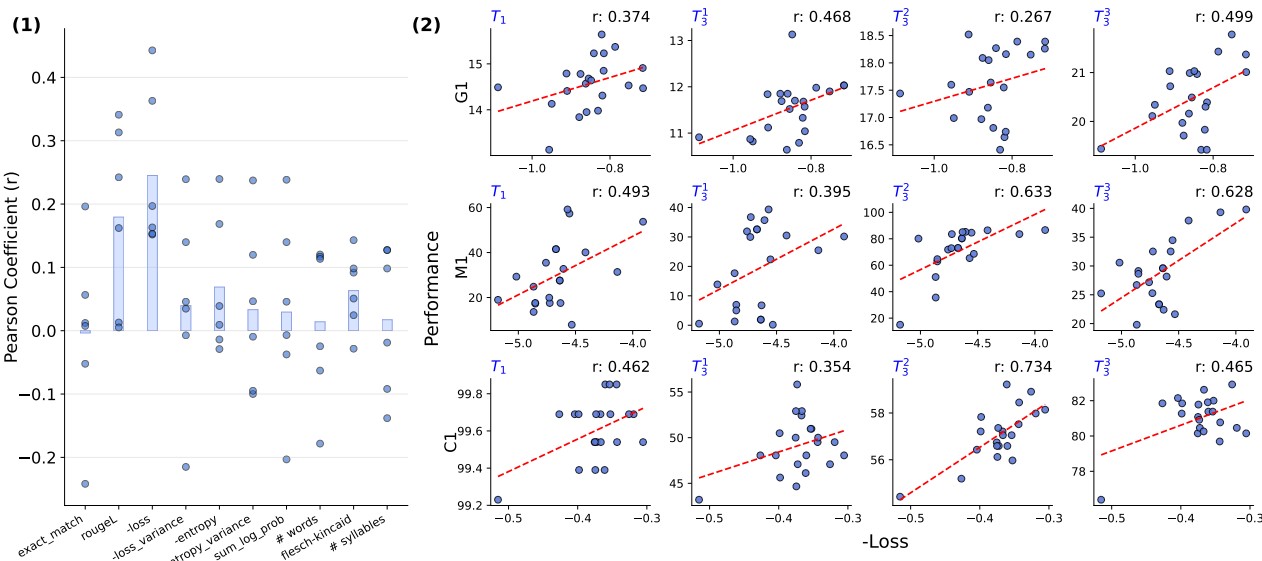

*Figure 12.* Pearson correlations between 10 pre-learning measurements and post-learning performance on other tasks. Results for Qwen3-8b over 120 task sequences.

Gradients are computed using the original, human-authored prompts. When calculating the gradients, we initialize new LoRA parameters identical to the standard training setup, while keeping all previous LoRA parameters frozen. However, during the gradient computation pass, we do not update any model parameters. This means the gradients for the lora_A matrices are zero, and we only analyze the gradients with respect to the lora_B parameters. Furthermore, we separately calculate the gradient angles between $T_2$ and $T_{1/3}$ for different modules. In our configuration, this corresponds to the lora_B parameters for the query (q) and value (v) matrices in each attention layer. Finally, based on prior work indicating that lower and middle layers of LLMs encode general knowledge while upper layers capture task-specific information (Meng et al., 2023; Zhao et al., 2024b), we restrict our statistical analysis of gradient angles to the model's upper layers. For the 32-layer Llama2-7b-chat model, this corresponds to the top 8 layers. Since these layers govern task-specific adaptation, their gradient alignment strongly suggests that the model is leveraging a shared solution pattern, effectively avoiding the formation of isolated, task-specific shortcuts.

## D. Analysis of Candidate Pool Size

In this section, we investigate the sensitivity of SAPO's performance to the size of the candidate prompt pool generated prior to training. First, we pre-generate a superset of 50 paraphrased prompts for each task in the sequence. Subsequently, immediately before training on any given task, we simulate varying candidate pool sizes $N$ (from 10 to 50) by constructing nested subsets from this superset. Specifically, to ensure consistency, the pool for a larger size (e.g., $N = 20$) strictly contains the entire subset used for the smaller size (e.g., $N = 10$). The optimal prompt, identified by the lowest pre-update loss within this designated subset, is then selected to guide the training. Figure 15 illustrates the performance trajectory as the candidate pool size increases. Experiments are conducted using the Llama2-7b-chat and Qwen3-8B model on NI-Seq-G1 and NI-Seq-C1 sequences, applying SAPO on top of O-LoRA. All reported results represent the average over two random seeds. We observe that with a small pool size ($N = 10$), performance gains are inconsistent, occasionally resulting in negligible improvement or even degradation compared to the baseline. In contrast, increasing the pool size to $N = 20$ yields consistent and stable performance boosts. Furthermore, expanding the pool beyond 20 candidates offers diminishing returns, with performance metrics plateauing. Consequently, we select a pool size of 20 as the standard setting for SAPO, representing an optimal trade-off between computational efficiency and performance maximization.

## E. Cost Analysis

In this section, we analyze the additional time overhead introduced by SAPO. As illustrated in Appendix D, SAPO evaluates candidates via forward passes on a small subset, introducing minimal overhead. Theoretically, fine-tuning a SuperNI task

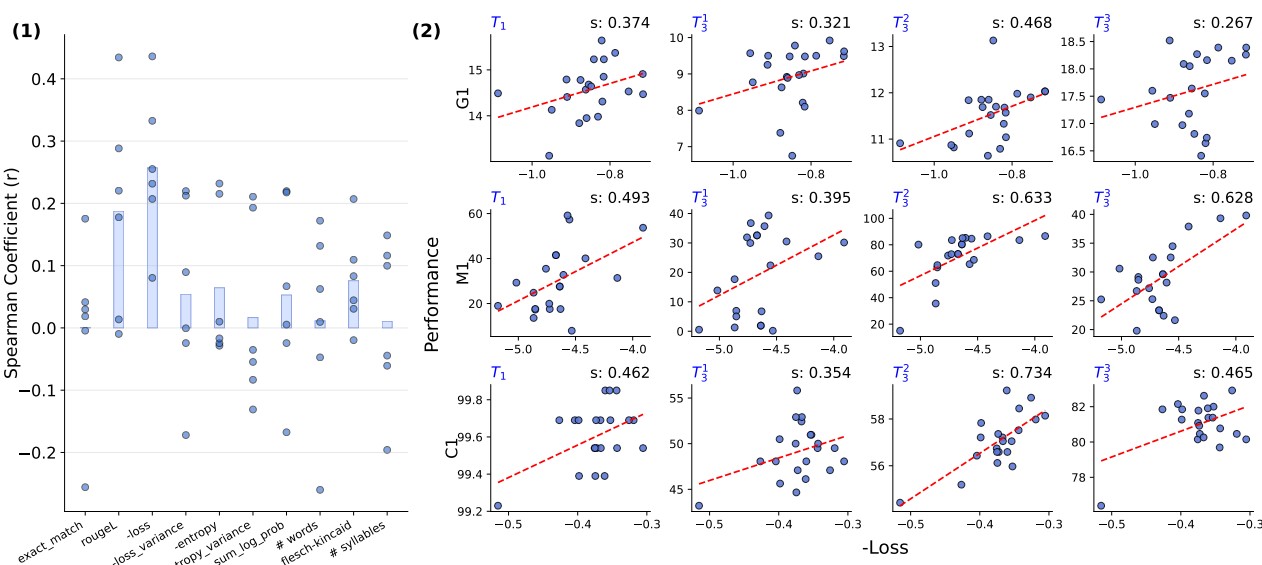

*Figure 13.* Spearman correlations between 10 pre-learning measurements and post-learning performance on other tasks. Results for Qwen3-8b over 120 task sequences.

requires 10,000 forward-backward passes (1,000 samples × 10 epochs), whereas SAPO needs only 5,000 forward passes (20 candidates × 250 samples). Since a forward-backward pass takes ∼3× more compute than a forward pass, the theoretical compute overhead of SAPO is merely ∼16.6%. Furthermore, gradient-free forward passes enable much larger batch sizes, significantly reducing wall-clock time. (Note that the time to generate paraphrases is negligible and excluded from this calculation).

Empirically, Table 10 quantifies the time for training Qwen3-8B on the NI-Seq-M1 sequence using 8 H20 GPUs. As an independent step, SAPO introduces a nearly constant and marginal time overhead (roughly 0.25 hours) regardless of the baseline. We will include the theoretical and empirical cost analysis in the revision.

## F. Supplementary Empirical Experiments

In Table 1, we compare our method against various baselines, evaluating performance across four models and four task sequences. To more robustly demonstrate the effectiveness of our approach, we provide supplementary results in Table 11, further detailing the performance of Llama2-7b-chat and Qwen3-8b on three additional SuperNI task sequences. In total, seven distinct task sequences are used to evaluate the methods in our empirical experiments. The specific composition of these sequences is illustrated in Table 9.

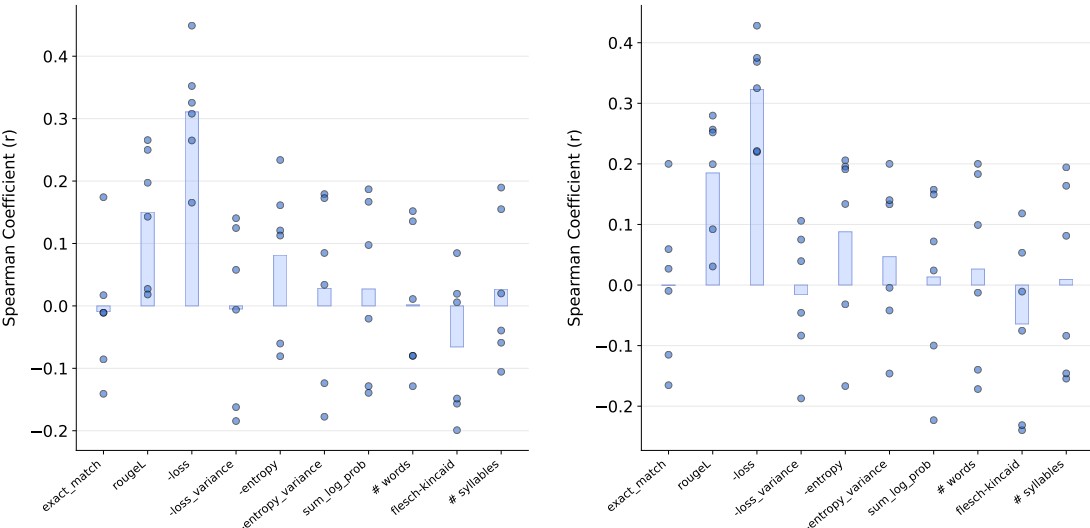

*Figure 14.* Pearson and Spearman correlations between 10 pre-learning measurements and post-learning performance on other tasks. Results for Qwen3-14b over 120 task sequences.

*Table 5.* Overview of the SuperNI dataset tasks.

| Dataset | Source | Category | Avg len | Metric | Language | #data |
|---------|--------|----------|---------|--------|----------|-------|
| NI002 | Quoref | Question Answering | 360 | ROUGE-L | English | 1000 |
| NI1290 | Xsum | Summarization | 363 | ROUGE-L | English | 1000 |
| NI1292 | Yelp review full | Sentiment Analysis | 130 | ROUGE-L | English | 1000 |
| NI141 | Odd man out | Word Semantics | 9 | ROUGE-L | English | 1000 |
| NI273 | Europarl | Text Matching | 15 | ROUGE-L | English | 1000 |
| NI024 | Cosmosqa | Question Answering | 82 | ROUGE-L | English | 1000 |
| NI1310 | Multilingual amazon reviews | Sentiment Analysis | 59 | ROUGE-L | English | 1000 |
| NI163 | Synthetic | Program Execution | 23 | ROUGE-L | English | 1000 |
| NI292 | Storycommonsense | Information Extraction | 48 | ROUGE-L | English | 1000 |
| NI1343 | Amazon us reviews | Sentiment Analysis | 70 | ROUGE-L | English | 1000 |
| NI195 | Sentiment140 | Sentiment Analysis | 14 | ROUGE-L | English | 1000 |
| NI1355 | Sentence compression | Summarization | 25 | ROUGE-L | English | 999 |
| NI589 | Amazon fine food reviews | Summarization | 84 | ROUGE-L | English | 1000 |
| NI1357 | Xlsum | Summarization | 454 | ROUGE-L | English | 1000 |
| NI360 | Numersense | Fill in The Blank | 26 | ROUGE-L | English | 1000 |
| NI339 | Record | Question Answering | 185 | ROUGE-L | English | 1000 |
| NI220 | Rocstories | Title Generation | 60 | ROUGE-L | English | 1000 |
| NI224 | Scruples | Ethics Classification | 338 | ROUGE-L | English | 1000 |
| NI611 | Mutual | Dialogue Generation | 162 | ROUGE-L | English | 1000 |
| NI1510 | Evalution | Information Extraction | 7 | ROUGE-L | English | 1000 |
| NI231 | Iirc | Question Answering | 229 | ROUGE-L | English | 1000 |
| NI488 | Synthetic | Program Execution | 16 | ROUGE-L | English | 1000 |
| NI618 | Multilingual amazon reviews | Summarization | 47 | ROUGE-L | English | 1000 |
| NI363 | Sst2 | Sentiment Analysis | 19 | ROUGE-L | English | 1000 |
| NI619 | Ohsumed | Title Generation | 161 | ROUGE-L | English | 1000 |
| NI511 | Reddit tifu dataset | Summarization | 400 | ROUGE-L | English | 1000 |

*Table 6.* Prompts for three marked points in Figure 5a.

| | Prompt |
|---|---|
| ■ | In this task, you're given a question, a context passage, and four options which are terms from the passage. After reading a passage, you will get a brief understanding of the terms. Your job is to determine by searching and reading further information of which term you can answer the question. Indicate your choice as 'a', 'b', 'c', or 'd'. If you think more than one option is plausible, choose the more probable option to help you answer the question. |
| ● | Read the question and the passage. Four key terms from the passage are labeled 'a', 'b', 'c', and 'd'. Determine which single term is most essential for answering the question. Output your selection as the corresponding lowercase letter. |
| ✕ | Given a question and a context passage with four labeled terms (a, b, c, d), identify which single term provides the necessary information to answer the question. Select the letter (a, b, c, or d) corresponding to the most relevant term. |

*Table 7.* Meta prompt used in Prompt Expansion to paraphrase the current-task prompt.

You will be given an instruction used for prompting a language model to perform a task.
Your job is to rewrite a **new instruction** that can guide a language model to perform the **same task**, but using a different style, structure, or tone.
Instruction: {cur_prompt}
Guidelines:
- The rewritten instruction should aim to achieve the same outcome or behavior as the original, but can use different words, length, structure, or phrasing.
- Creativity is encouraged, as long as the instruction is still suitable for the same task.
- If the original prompt includes any task labels (e.g., "Positive", "Negative"), **they must be preserved exactly**, including spelling and case. - Do not mention that this is a paraphrase.
- Output your rewritten instruction between $<$ START $>$ and $<$ /START $>$.

*Table 8.* A summary of dataset statistics in TRACE benchmark.

| Dataset | Source | Category | Avg len | Metric | Language | #data |
|---|---|---|---|---|---|---|
| ScienceQA | Science | Multi-Choice QA | 210 | ROUGE-L | English | 3,000 |
| FOMC | Finance | Multi-Choice QA | 51 | ROUGE-L | English | 3,000 |
| MeetingBank | Meeting | Summary | 2853 | ROUGE-L | English | 3,000 |
| C-STANCE | Social media | Multi-Choice QA | 127 | ROUGE-L | Chinese | 3,000 |
| Py150 | Github | Code generation | 422 | ROUGE-L | Python | 3,000 |
| NumGLUE-cm | Math | Math reasoning | 32 | ROUGE-L | English | 3,000 |

*Table 9.* Information of continual learning task sequences used in empirical experiments.

| | Sequence | Task Type | Num. per Task |
|---|---|---|---|
| NI-Seq-G1 | NI589 → NI141 → NI618 → NI339 → NI360 | Generation | 1,000 |
| NI-Seq-G2 | NI589 → NI024 → NI360 → NI511 → NI618 | Generation | 1,000 |
| NI-Seq-C1 | NI195 → NI1310 → NI273 → NI611 → NI224 | Generation | 1,000 |
| NI-Seq-C2 | NI231 → NI195 → NI1292 → NI224 → NI363 | Generation | 1,000 |
| NI-Seq-M1 | NI195 → NI360 → NI611 → NI002 → NI224 | Cls. & Gen. | 1,000 |
| NI-Seq-M2 | NI618 → NI195 → NI360 → NI363 → NI589 | Gen. & CLS | 1,000 |
| TRACE | C-Stance → Fomc → Meet → Py150 → SciQA → Numgluecm | Mixed | 3,000 |

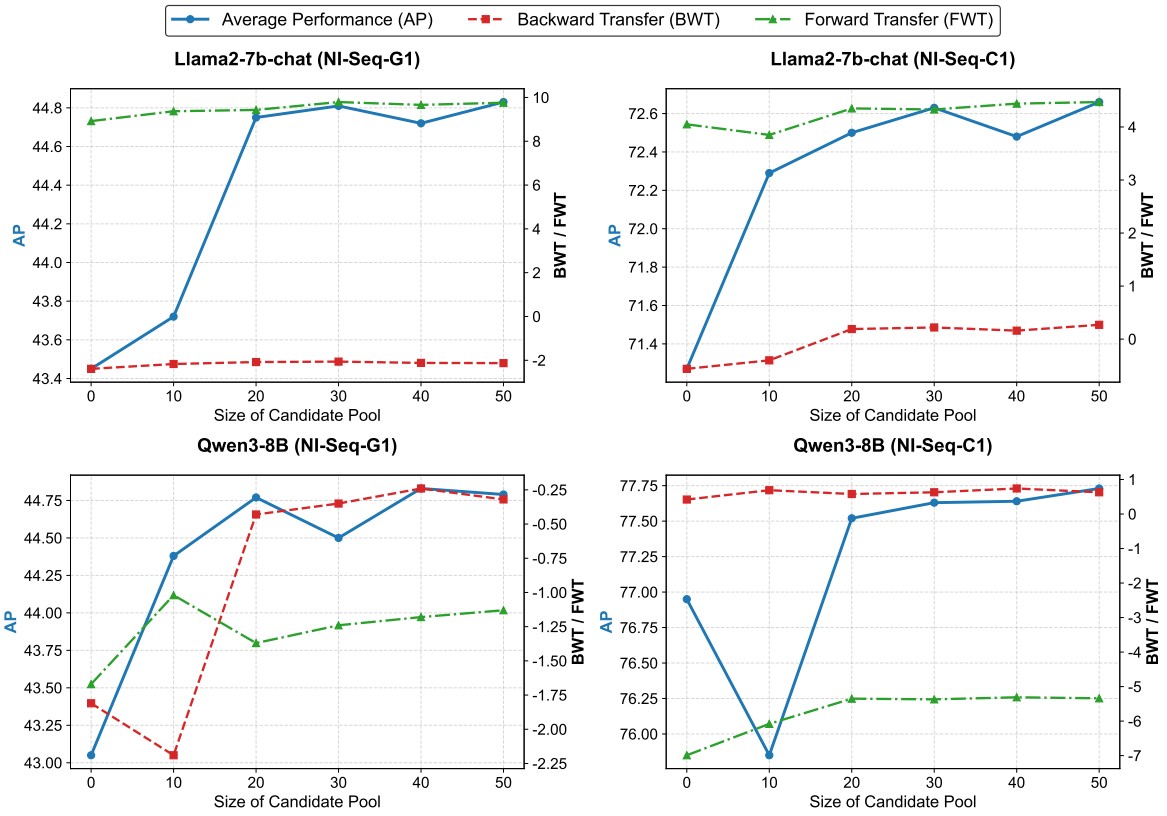

*Figure 15.* Impact of candidate pool size on SAPO performance. The curves depict the performance on NI-Seq-G1 and NI-Seq-C1 using Llama2-7b-chat and Qwen3-8B equipped with O-LoRA.

*Table 10.* Empirical training time analysis of Qwen3-8B on the NI-Seq-M1 sequence using 8 H20 GPUs.

| Baseline | Original Time (h) | + SAPO Time (h) | Overhead |
|----------|-------------------|-----------------|----------|
| LoraInc  | 2.0               | + 0.25          | 12.5%    |
| O-Lora   | 2.2               | + 0.25          | 11.4%    |
| InsCL    | 2.5               | + 0.25          | 10.0%    |
| EWC      | 3.0               | + 0.25          | 8.3%     |

*Table 11.* Performance of baselines and their improved version with SAPO on additional three benchmarks.

| | Method | NI-Seq-G2 | | | NI-Seq-C2 | | | NI-Seq-M2 | | |
|---|---|---|---|---|---|---|---|---|---|---|
| | | AP ↑ | BWT ↑ | FWT ↑ | AP ↑ | BWT ↑ | FWT ↑ | AP ↑ | BWT ↑ | FWT ↑ |
| **Llama2-7b-chat** | LoraInc | 22.78 | -5.84 | 0.33 | 83.38 | -0.31 | 11.35 | 41.8 | -7.72 | 0.66 |
| | +SAPO | **+0.65** | **+0.88** | **+1.21** | **+0.13** | **+0.11** | **+4.57** | **+0.5** | **+1.17** | **+0.3** |
| | EWC | 24.35 | -2.14 | 0.12 | 81.32 | -4.63 | 16.82 | 41.93 | -14.55 | 2.75 |
| | +SAPO | **+0.67** | **+0.33** | **+0.31** | **+1.17** | **+1.9** | **-1.79** | **+1.33** | **+2.15** | **+4.11** |
| | O-Lora | 24.93 | -1.64 | 1.61 | 82.88 | -0.51 | 9.05 | 43.75 | -3.02 | -0.25 |
| | +SAPO | **+0.54** | **+0.35** | **+0.84** | **-0.35** | **+0.43** | **+0.01** | **+1.24** | **+0.84** | **+2.58** |
| | InsCL | 24.94 | -0.56 | 0.63 | 83.81 | 1.21 | 13.16 | 50.05 | -0.79 | 0.39 |
| | +SAPO | **+0.24** | **+0.45** | **+1.55** | **+1.54** | **+3.05** | **+0.61** | **+1.02** | **+0.91** | **+2.36** |
| **Qwen3-8b** | LoraInc | 23.13 | -2.01 | 0.10 | 83.39 | -0.87 | -3.16 | 42.32 | -4.15 | -0.13 |
| | +SAPO | **+0.57** | **+0.49** | **+-0.03** | **+0.5** | **+0.31** | **+1.32** | **+0.23** | **+0.6** | **+0.12** |
| | EWC | 25.08 | -0.30 | -0.34 | 83.06 | -0.45 | -3.29 | 44.36 | -1.12 | -0.28 |
| | +SAPO | **+0.32** | **+0.45** | **+1.15** | **+1.13** | **+0.85** | **+1.61** | **+0.31** | **+0.65** | **-0.15** |
| | O-Lora | 24.87 | 0.05 | -0.42 | 82.61 | -0.27 | -1.22 | 44.29 | -0.89 | -0.43 |
| | +SAPO | **+0.67** | **+0.93** | **+1.62** | **+0.58** | **+0.36** | **+0.57** | **+0.36** | **+0.85** | **+0.29** |
| | InsCL | 26.45 | 0.09 | -0.16 | 84.78 | -0.34 | -2.98 | 45.49 | -0.53 | 0.16 |
| | +SAPO | **+0.59** | **+0.41** | **+1.04** | **+0.46** | **+0.49** | **+1.30** | **+0.90** | **+0.22** | **+0.20** |

