# OpenReview forum: "Training Prompt Matters: State-Adaptive Optimization for Robust Fine-Tuning"
_ICML.cc/2026/Conference — ICML 2026 regular_

### Official Review · Reviewer_Ttut · 2026-03-05

**Soundness:** 2
**Presentation:** 2
**Significance:** 2
**Originality:** 3
**Overall Recommendation:** 3
**Confidence:** 3

**Summary:**

The manuscript shows that the prompt used in fine-tuning can affect other task performance in learning a sequence of tasks. The authors propose to the importance of selecting/computing an optimal prompt based on the task loss to induce better generalizability and less forgetting.

**Compliance With Llm Reviewing Policy:**

Affirmed.

**Final Justification:**

I still hold my previous comments. The study is sound, but I also find it limited in its experimental setup, new insights, and potential impact.

**Key Questions For Authors:**

The authors claimed the prompts are paraphrased, and the difference is only lexical and syntactic. However, in the only example given in the main tex: “Create a concise summary based on the provided Amazon product review” v.s. “Using Amazon’s products reviews provided, create a Summary of the review”, there is a clear semantic difference. The first one contains “concise” which the second one doesn’t. This could affect the downstream performance in a way that the meaning of “concise” is messed up, especially when the other tasks are about conciseness.

Related to my previous point, there is only few details about the dataset and samples. To truly appreciate the contribution of the claim of this manuscript, it is important to know what the applicable domains are. The tasks are from a general NLP dataset (SuperNI). It looks like the authors picked dozens of tasks from this dataset and classified them into three types (generation, classification, mixed). There are surely correlations between the tasks (e.g. maybe multiple tasks are about summarization, but in different styles). Without examples and details, it is hard to interpret those “observations”. For example, the generalization and forgetting are observed to be coupled, is that because the tasks are overall similar to each other? Then it doesn’t explain why on the training tasks, there is significant smaller variance between prompts. Questions like this are not answered in the manuscript.

3.3 need clarification. It looks like the main claim is that there are prompts that generalize well on multiple tasks. But the resulting plot can be interpreted in a different way: there are tasks that are highly correlated (Figure 2). If the tasks are similar enough, it is trivial that one model does well on on task will also does well on the correlated tasks. Is it that the model is universally better, or the tasks are too similar to each other.

Minor: line 208 “Our findings advance the mechanistic understanding of finetuning by elucidating the pivotal role of training prompts.” These findings are not “mechanistic”. Though these terminologies are somewhat arbitrary in current AI literature, “mechanistic” usually means analyzing the model's internal activations.

One emphasized advantage of SAPO is “state-adaptive”. It matches a widespread understanding that using model’s intrinsic distribution causes less forgetting. But there is an alternative hypothesis: the low loss may not depend on the “state”, but be constant, such that computing the optimal prompt from the base model could perform as well. The manuscript lacks a control/ablation for SAPO that is the non-state-adaptive version, for which the optimal prompt is computed using a fixed model.

**Limitations:**

Yes.

**Strengths And Weaknesses:**

Strength: The fine-tuning prompt is an understudied topic, and the study provided some preliminary insights.

Weakness: However, the experiments are limited to sequentially learning classic NLP-style tasks. From my personal view, this setup is rarely reflected in the real-world LLM post-training scenarios, and makes some claims questionable as detailed below.

---

> ### Author Rebuttal · Authors · 2026-03-30
>
> We sincerely thank the reviewer for the constructive and insightful feedback.
>
> **Weakness**: We agree that LLM post-training often involves single-task fine-tuning rather than sequential learning. However, single-task learning still faces the exact same fundamental challenges of catastrophic forgetting and cross-task generalization.
> Continual learning setup explicitly isolates these two phenomena, allowing us to precisely analyze how training prompts affect both forgetting and generalization. Furthermore, sequential learning is inherently more difficult and encompasses single-task learning. Since SAPO consistently improves performance across a challenging sequence of multiple tasks, its effectiveness naturally extends to single-task SFT.
>
> **Q1**: We thank the reviewer for pointing out this oversight and agree that our original phrasing was inaccurate. While the majority of our paraphrases involve only lexical and syntactic changes, minor semantic variations (e.g., adding or omitting slight constraints) do occasionally occur.
> However, our core finding strictly holds: the choice of training prompts heavily impacts cross-task generalization, and pre-update loss reliably identifies superior prompts.
> In addition, even when prompts are strictly semantically identical, they still yield notably different cross-task performances. For instance, the following prompts from Figure 1(c) share the exact same semantics but still led to significant performance variances after training. In the revision, we will correct our wording to accurately reflect these prompt variations, clarify our conclusions, and include more examples to further demonstrate the impacts of training prompts.
>
> "Given an Amazon product review, create a summary of the review."
>
> "Generate a summary for the given Amazon product review."
>
> "You are given an Amazon product review. Your task is to produce a summary of the review."
>
> **Q2**: We apologize for the limited details regarding our experimental setup. We selected 26 diverse tasks from SuperNI, covering various types (e.g., sentiment analysis, question answering, summarization, program execution) and difficulty levels, as detailed in Table 4.
> We categorized these tasks into classification and generation types to construct three distinct sequences: pure classification, pure generation, and mixed (alternating). These sequences inherently possess varying degrees of correlation. For instance, the pure classification sequence exhibits significantly less forgetting than the others, indicating stronger inter-task correlation.
> Crucially, our core observations hold robustly across all sequence types. Whether tasks are strongly correlated (classification), weakly correlated (generation), or lack obvious correlation (mixed), the training prompt heavily impacts cross-task performance, with generalization and forgetting consistently coupled. This demonstrates that our findings are not merely artifacts of overall task similarity.
> Furthermore, we validate our method on the TRACE benchmark, which includes even more diverse and heterogeneous tasks (e.g., Science QA, mathematical reasoning, code generation), further substantiating our findings.
>
> **Q3**: Regarding Section 3.3, Figure 2 presents both pure generation and mixed. The cross-task performance is all evaluated on a highly diverse set of generation tasks, such as question answering, summarization, program execution, title generation, and dialogue generation.
> These tasks involve fundamentally different requirements and linguistic structures. This diversity confirms that superior training prompts indeed make the model universally better, rather than the evaluation tasks merely being highly correlated.
> We will include a more detailed introduction of these tasks in the revision to fully clarify our findings and demonstrate the broad applicability of our method.
>
> **Q4**: We thank the reviewer for this correction. We agree that the term "mechanistic" is typically reserved for the analysis of internal activations and will correct this terminology in the revision to provide a more accurate description of our findings.
>
> **Q5**: To isolate the value of the "state-adaptive" mechanism, we compare SAPO against a non-state-adaptive baseline (PO).
> As shown in the table below (using Qwen3-8B on the NI-Seq-M1 sequence, averaged over 3 runs), SAPO achieves consistent improvements across all metrics, whereas PO fails to guarantee performance gains. A fine-grained analysis reveals that PO only provides consistent benefits on the first task, demonstrating that the optimal prompt is not constant, and dynamic, state-adaptive prompt selection is indeed necessary. We will include more comprehensive experiments and analyses in the revision to highlight the value of SAPO's state-adaptive nature.
> |Baseline|Method|Δ AP|Δ BWT|Δ FWT|
> |:---|:---|:---|:---|:---|
> |LoraInc|+SAPO|+1.89|+0.50|+0.06|
> ||+PO|-0.15|-0.36|+0.13|
> |O-Lora|+SAPO|+1.07|+0.53|+2.97|
> ||+PO|-0.23|+0.18|-0.42|

---

> > ### Author Rebuttal · Reviewer_Ttut · 2026-04-03
> >
> > I would like to thank the authors for the clarification. What the authors did is sound. However, I am still concerned about the 1) semantic changes in the prompt, 2) that fine-tuning prompt is constrained to SuperNI tasks, which are not the majority of the LLM pre-training and post-training focus, 3) the SAPO method requires randomly sampling prompts, and the Pre-Update Loss as a predictor lacks deeper mechanistic understanding. I can imagine a gradient based optimization method for optimal prompt would perform better; 4) the effects of fine-tuning prompt are interesting, but this study only probed it at a superficial level in a constrained setup.
> >
> > I will keep my score.

---

> > > ### Author Response · Authors · 2026-04-04
> > >
> > > We thank the reviewer for the helpful feedback. We are glad that some concerns have been clarified, and we briefly address the remaining ones below.
> > >
> > > (1) **On semantic changes in paraphrases.** We agree that not all paraphrases are perfectly semantics-preserving. Most variations are lexical or syntactic, and when semantic changes do exist, they are typically limited to minor constraints. Importantly, however, our main conclusion does not rely on perfect semantic identity. The key finding is that even small changes in training prompts can induce large and systematic differences in cross-task behavior, and these differences are predictable from the pre-update loss. This is supported not only by the probing results, but also by the fact that SAPO consistently improves performance across models, baselines, and benchmarks, while deliberately selecting high-loss prompts consistently degrades performance. Therefore, whether the variation is purely lexical or includes slight semantic drift, the main conclusion remains the same: training prompt formulation significantly affects post-training behavior and is an optimizable variable.
> > >
> > > (2) **On the scope beyond SuperNI.** We use SuperNI because it provides a controlled yet diverse testbed for systematically studying the effect of training prompts. This controlled setup is important for isolating the relationship between prompt choice, forgetting, and generalization. At the same time, the paper is not limited to SuperNI. We also evaluate on TRACE, which includes science QA, mathematical reasoning, code generation, settings that are much closer to current LLM post-training practice. In this sense, the controlled SuperNI analysis and the broader TRACE evaluation serve complementary roles: the former establishes the phenomenon clearly, while the latter shows that the method remains effective in more heterogeneous and practically relevant settings.
> > >
> > > (3) **On candidate generation and mechanistic depth.** We agree that SAPO is not the only possible optimization strategy, and gradient-based prompt optimization is a promising future direction. However, this does not weaken the contribution of the paper. Our key contribution is to identify training prompt formulation itself as a new optimization target and to show that even a simple state-adaptive selection strategy already yields reliable gains with minimal overhead. The paraphrase pool is only a practical instantiation of this idea, not the core claim itself.
> > > Moreover, our claim does not rely only on predictive correlation. In Section 5.4, we further conduct mechanism analysis and show that lower-loss prompts lead to better gradient alignment between the current task and non-training tasks, indicating reduced optimization conflict. This provides mechanistic evidence for why lower-loss prompts help mitigate forgetting while improving generalization.
> > >
> > > (4) **On the depth of the study.**
> > > Our study focuses on paraphrase-level prompt variations, which may appear superficial. However, this is precisely why the finding is important: even such minimal variations in training prompts can substantially affect post-training behavior, suggesting that the impact of prompt formulation may be even more pronounced for more complex or semantically diverse prompts. Starting from this "superficial" setting, wewe systematically investigate training prompt effects and reveal that prompt selection is an important yet previously overlooked factor in fine-tuning. We believe this provides a necessary foundation for studying richer forms of training prompt variation in the future.
> > > Moreover, our constrained continual-learning setup is intentional: it allows us to clearly disentangle the effect of training prompts on forgetting and generalization. Concretely, we select 26 diverse SuperNI tasks, construct 120 task sequences, and evaluate across four model families/scales. Beyond the probe study, we further conduct 72 empirical experiments on SuperNI and TRACE, covering multiple task sequences, baselines, and models, where SAPO consistently improves performance. We believe this level of systematic evaluation provides robust evidence that training prompt selection is an important and practical factor in fine-tuning.
> > >
> > > We hope these responses clarify our position and make the contribution of the paper clearer.

---

### Official Review · Reviewer_x5Yf · 2026-03-08

**Soundness:** 3
**Presentation:** 3
**Significance:** 2
**Originality:** 3
**Overall Recommendation:** 4
**Confidence:** 4

**Summary:**

This paper investigates the critical role of training prompts during the fine-tuning of LLMs. Existing fine-tuning paradigms typically assume that as long as prompts are semantically equivalent, the model's learning outcomes will be consistent. However, the authors astutely observe that this "semantic equivalence" is highly deceptive. Research indicates that while semantically similar variants converge in performance on the current fine-tuning task, they induce drastically different and profound shifts in the model's global knowledge distribution. Based on the empirical finding that "Pre-update Loss is highly correlated with cross-task performance," the paper proposes SAPO. This method transforms static prompt inputs into dynamic variables tailored to the model's current parameter state.

**Compliance With Llm Reviewing Policy:**

Affirmed.

**Final Justification:**

Overall, the authors successfully demonstrated the interesting and counterintuitive phenomenon that ``the choice of training cues has a significant impact on model state and catastrophic forgetting,'' and their proposed method is simple and effective. Given that my main technical concerns have been clarified, I have decided to raise my score to 4.

**Key Questions For Authors:**

1. When calculating the pre-update loss, is it computed over the entire sequence or strictly over the answer portion? In standard instruction tuning, gradients are typically only calculated for the Answer. If the Loss computation includes the Instruction portion, does it introduce distribution noise irrelevant to the actual task learning?

2. The paper mentions that the 20 generated variants must "strictly match the length of the original prompt." Operationally, how is this "match" defined? Is it an identical token count, or is it allowed to fluctuate within a specific percentage error margin?

3. The geometric analysis suggests that low-loss prompts reduce gradient conflicts. Does this imply that SAPO inherently favors phrasings that the model has "already mastered best during the pre-training phase"? If this inference holds true, when a fine-tuning task involves completely "new knowledge" unfamiliar to the model, can Pre-update Loss still serve as a reliable prior indicator for predicting cross-task performance?

4. SAPO conducts evaluation and selection only once before each new task begins. However, within the training cycle of a single task, the model's weights are continuously updated, causing its "internal state" to shift. Is the initially selected prompt still the optimal solution in the middle to late stages of training?

**Limitations:**

yes

**Strengths And Weaknesses:**

Strengths：

1. The paper successfully identifies and quantifies a hidden variable—the "alignment degree between fine-tuning prompts and model states"—challenging the traditional assumption that "training prompts are merely surface forms." This offers a novel perspective for enhancing the robustness of LLM fine-tuning.

2. The SAPO strategy is logically sound, plug-and-play, and provides a reasonable theoretical justification through geometric analysis for utilizing pre-update loss as a filtering metric.

Weaknesses：

1. The effectiveness of SAPO heavily relies on the 20 candidate prompts generated by an external model (e.g., Gemini). Would the gains significantly diminish if the generated prompts were of low quality or lacked diversity? The paper lacks an ablation study regarding the quality of the candidate pool to demonstrate the method's robustness.

2. The paper focuses on finding a "unique" prompt with the lowest loss during training, which might lead the model to overfit to specific phrasings. Since users typically employ "standard phrasings" during inference, could this training-side local optimization degrade the model's responsiveness to standard inference prompts?

3. The current experiments predominantly focus on traditional instruction/task-oriented datasets like SuperNI and TRACE. The applicability of SAPO to more core capabilities of modern LLMs, such as "multi-turn dialogue" or "complex reasoning," remains unverified, thereby limiting its universality to some extent.

---

> ### Author Rebuttal · Authors · 2026-03-30
>
> We are grateful for the reviewer’s constructive and insightful comments.
>
> **W1**: Our work reveals that even semantically consistent paraphrases yield significant impacts, and SAPO effectively identifies the optimal variant to achieve notable gains.
> Since SAPO-selected prompts involve only simple lexical or structural variations, which are trivial for modern LLMs to generate, it does not demand high prompt quality.
> The table below reports averaged results with 3 runs on Qwen3-8B (NI-Seq-M1) using Gemini-2.5-Pro, GPT-OSS-120B, and Qwen3-32B as paraphrasers. SAPO achieves consistent improvements across all models, confirming its robustness.
> |Baseline|Paraphraser|Δ AP|Δ BWT|Δ FWT|
> |:---|:---|:---|:---|:---|
> |LoraInc|Gemini|+1.89|+0.50|+0.06|
> ||GPT|+1.47|+1.21|+0.31|
> ||Qwen|+1.66|+0.62|+0.05|
> |O-Lora|Gemini|+1.07|+0.53|+2.97|
> ||GPT|+1.11|+0.47|+1.79|
> ||Qwen|+0.55|+0.76|+2.49|
>
> Regarding prompt diversity, a pool of 20 candidates provides sufficient variance. SAPO achieves consistent success across 72 diverse experiments (Tables 1 and 9) using just 20 candidates, demonstrating its diversity requirement is easily satisfied in practice. Further ablation analysis is in Appendix D, showing that smaller pools can diminish improvements.
> We will further clarify SAPO's requirements for candidate pool quality and diversity in the revision.
>
> **W2**: To clarify, SAPO identifies the lowest-loss prompt before training. And after training, all paraphrased prompts converge to similar losses with comparable in-task performance. But the initial low-loss prompts achieve better cross-task performance, indicating they actually mitigate overfitting rather than exacerbate it.
> Regarding inference mismatch, we train Qwen3-8B on Tasks 195 and 231 using four prompts ranked by pre-update loss, and evaluate with both matched and mismatched prompts. As shown below, while mismatched prompts perform worse, this performance drop does not correlate with the pre-update loss ranking, indicating SAPO introduces no additional adverse effects. We will include relevant analysis in the revision to further clarify the potential impact of SAPO.
> | Prompt | T195 (Matched) |T195 (Mis-M)|T231 (Matched)|T231 (Mis-M)|
> |:---|:---|:---|:---|:---|
> |P1 (Lowest loss)|83.8|82.5| 89.7|88.4|
> |P2|83.7|82.1|89.9|89.3|
> |P3|84.1|82.4|89.3|87.8|
> |P4 (Highest)|83.7|81.6|89.5|87.7|
>
> **W3**: Although SuperNI is instruction-oriented in formulation, it covers a broad spectrum of task types and difficulties (e.g., question answering, summarization, program execution) and is widely adopted in recent continual learning literature [1, 2]. Furthermore, our evaluation on TRACE benchmark includes complex tasks like mathematical reasoning and code generation, which heavily rely on the core capabilities of modern LLMs.
> Regarding multi-turn dialogue, since SAPO optimizes explicit task descriptions or instructions, it is difficult to apply to open-ended dialogue tasks that lack these structured elements.
> We will clarify the evaluated task types and our method's limitations in the revision.
>
> **Q1**: As detailed in Section 4.1 (Model-behavior signals), the pre-update loss is computed as the causal language modeling loss over the target outputs.
>
> **Q2**: We thank the reviewer for pointing this out and acknowledge that "strictly match" is an overstatement. While our original intention is to control length to demonstrate the model's sensitivity to prompt phrasing, the generated paraphrases only roughly approximate the original length in practice. We will correct this wording to accurately reflect the approximate length matching in the revision.
>
> **Q3**: We agree that SAPO favors phrasings the model better mastered during pre-training. As Section 5.4 validates, lower-loss training prompts reduce specific adaptations and mitigate potential knowledge conflicts.
> For tasks that contain genuinely novel knowledge, the model still needs to learn other partially familiar information. Therefore, low-loss prompts remain highly beneficial: by minimizing the need to learn specific stylistic or structural adaptations, they reduce knowledge conflicts and allow the model to focus more on acquiring new knowledge. Thus, pre-update loss is a reliable prior indicator even for unfamiliar tasks.
>
> **Q4**: The initially selected prompt remains optimal throughout training. It starts with the lowest loss, and our loss trajectory observations show its loss decreases the fastest among all paraphrases, which is intuitive as model is explicitly optimized on this exact prompt.
> This dynamic also explains our inference mismatch observation: non-training prompts cannot achieve the low loss of the actively trained prompt, naturally leading to worse performance.
>
> [1] Unlocking the power of function vectors for characterizing and mitigating catastrophic forgetting in continual instruction tuning. ICLR 2025
>
> [2] Recurrent Knowledge Identification and Fusion for Language Model Continual Learning. ACL 2025

---

> > ### Author Rebuttal · Reviewer_x5Yf · 2026-04-03
> >
> > Thank you to the authors for their detailed response and the supplementary experiments and explanations provided during the rebuttal. Your response addressed most of my concerns. Overall, the authors successfully demonstrated the interesting and counterintuitive phenomenon that ``the choice of training cues has a significant impact on model state and catastrophic forgetting,'' and their proposed method is simple and effective. Given that my main technical concerns have been clarified, I have decided to raise my score to 4.

---

> > > ### Author Response · Authors · 2026-04-04
> > >
> > > We are very pleased that our response has clarified your concerns, and we sincerely thank you for your encouraging feedback and for your willingness to raise your score. Your thoughtful comments and suggestions have been very helpful in improving the quality and presentation of our paper. Based on your suggestions, we will carefully revise the manuscript and incorporate the additional experimental results and related discussion into the final version. We are deeply grateful for the time and effort you have devoted to reviewing our work and for your valuable feedback and encouragement throughout the process.

---

### Official Review · Reviewer_EerM · 2026-03-10

**Soundness:** 3
**Presentation:** 3
**Significance:** 2
**Originality:** 2
**Overall Recommendation:** 4
**Confidence:** 3

**Summary:**

Although semantically equivalent prompt paraphrases often lead to similar in-task performance, they can produce markedly different cross-task effects in terms of catastrophic forgetting and generalization. Moreover, this variation appears to be systematic rather than random, and the paper suggests that better prompts can be identified through pre-update task loss. Based on this observation, this paper propose State-Adaptive Prompt Optimization (SAPO), a lightweight method that dynamically adapts task instructions to the model’s evolving training state. Extensive experiments across multiple benchmarks indicate that SAPO mitigates forgetting and improves generalization, outperforming prior continual fine-tuning baselines. Overall, the paper highlights training prompt design as an important factor shaping learning dynamics and offers a practical method for more robust fine-tuning.

**Compliance With Llm Reviewing Policy:**

Affirmed.

**Final Justification:**

This paper propose State-Adaptive Prompt Optimization (SAPO), a lightweight method that dynamically adapts task instructions to the model’s evolving training state. Considering that this paper offers inspirational value to the research community, and the authors have addressed my concerns to some extent in the rebuttal, I will adjust my score to 4.

**Key Questions For Authors:**

Please refer to the Weakness.

**Limitations:**

yes

**Strengths And Weaknesses:**

Strengths:
1. The paper is well written and easy to follow; the exposition is fluent and the terminology is used precisely.
2. The proposed method is simple yet effective, easy to plug into existing pipelines, and largely orthogonal to existing approaches.
3. The experimental design is fairly comprehensive and, overall, provides reasonable support for the main claims of the paper.

Weaknesses:

1. SAPO relies on external models for prompt rewriting. It remains unclear how robust the method is to different rewriting models or paraphrasing strategies. In addition, the paper could better clarify how paraphrase quality is objectively assessed.
2. It would be helpful to include a workflow figure to present the SAPO algorithm more intuitively. In addition, the cost of SAPO is not sufficiently clear and should be quantified more explicitly. Some writing details also need attention; for example, in Lines 298–299, Gemini should be specified with an exact version number, and the caption of Figure 2 uses “LLama2” whereas the rest of the paper uses “Llama2.”
3. All experiments are conducted in a continual learning setting, where the model learns tasks sequentially. However, the core insight of the paper—that training prompts affect cross-task behavior—may also hold in broader fine-tuning scenarios. It would strengthen the paper to discuss whether SAPO can generalize to more standard SFT or domain adaptation settings beyond continual learning.
4. In some settings, the gains of SAPO appear relatively limited, and many task pairs show only moderate Pearson correlations (around 0.3–0.6). This may suggest that the relationship between loss and cross-task performance is fairly noisy. If so, it would be helpful for the authors to clarify when this proxy is reliable and when it may lead to suboptimal prompt selection.

---

> ### Author Rebuttal · Authors · 2026-03-30
>
> We sincerely thank the reviewer for the valuable feedback.
>
> **W1**: Our work reveals that even semantically consistent paraphrases of training prompts yield significant impacts, and SAPO effectively identifies the optimal variant. Because generating these simple variations is trivial for modern LLMs, SAPO does not rely on special rewriting models or paraphrasing strategies. The table below reports averaged results with 3 runs on Qwen3-8B (NI-Seq-M1) using Gemini-2.5-Pro, GPT-OSS-120B, and Qwen3-32B as paraphrasers. SAPO achieves consistent gains across all models, confirming its robustness.
> |Baseline|Paraphraser|Δ AP|Δ BWT|Δ FWT|
> |:---|:---|:---|:---|:---|
> |LoraInc|Gemini|+1.89|+0.50|+0.06|
> ||GPT|+1.47|+1.21|+0.31|
> ||Qwen|+1.66|+0.62|+0.05|
> |O-Lora|Gemini|+1.07|+0.53|+2.97|
> ||GPT|+1.11|+0.47|+1.79|
> ||Qwen|+0.55|+0.76|+2.49|
>
> To objectively assess paraphrase quality, we design three types of quantitative metrics: prompt-intrinsic (e.g., word count, syllable count), model-behavior (e.g., loss, Rouge-L), and uncertainty. By evaluating the correlation between these metrics and post-training performance across 120 task sequences, we empirically find the low-loss training prompt is optimal.
> Furthermore, SAPO requires paraphrase diversity, which can be reflected by the candidate pool size. As detailed in Appendix D, a pool size of 20 yields consistent gains, whereas smaller pools produce inconsistent results. We will include a comprehensive analysis of rewriting model and paraphrase quality in the revision.
>
> **W2**: We appreciate the constructive suggestions. In the revision, we will add a workflow figure to illustrate the SAPO algorithm and correct the model names to Gemini-2.5-Pro and Llama2.
> Regarding computational cost, as illustrated in Appendix C.4, SAPO evaluates candidates via forward passes on a small subset, introducing minimal overhead. Theoretically, fine-tuning a SuperNI task requires 10,000 forward-backward passes (1,000 samples × 10 epochs), whereas SAPO needs only 5,000 forward passes (20 candidates × 250 samples). Since a forward-backward pass takes ~3× more compute than a forward pass, SAPO's theoretical compute overhead is merely ~16.6%. Furthermore, gradient-free forward passes enable much larger batch sizes, significantly reducing wall-clock time. (Note taht the time to generate paraphrases is negligible and excluded from this calculation).
> Empirically, the table below quantifies the time for training Qwen3-8B on NI-Seq-M1 sequence using 8 H20 GPUs. As an independent step, SAPO introduces a nearly constant and marginal time overhead (roughly 0.25 hours) regardless of the baseline. We will include the theoretical and empirical cost analysis in the revision.
> ||Original Time + SAPO Time (Overhead %)|
> |:---|:---|
> |LoraInc|2.0 + 0.25 (12.5 %)|
> |O-Lora|2.2 + 0.25 (11.4 %)|
> |InsCL|2.5 + 0.25 (10.0 %)|
> |EWC|3.0 + 0.25 (8.3%)|
>
> **W3**: We utilize continual learning as a controlled setting to distinguish between catastrophic forgetting and generalization, which are difficult to disentangle in standard SFT or domain adaptation. By isolating these factors, we can more precisely illustrate the cross-task impact of training prompts.
> Furthermore, each learning task in our continual sequence is fundamentally an independent, standard SFT.
> Since Our reported empirical metrics represent the average performance changes across these consecutive SFT steps, SAPO's consistent overall improvements indicate that it successfully enhances cross-task behavior in the vast majority of individual SFT instances.
> Additionally, our evaluation covers diverse task types (e.g., question answering, sentiment analysis, mathematical reasoning) and domains (e.g., science, finance, social media), demonstrating that SAPO generalizes well to broader settings. We will include supplementary single-task fine-tuning results in the revision to further validate SAPO's broad applicability.
>
> **W4**: We acknowledge that some task pairs exhibit moderate Pearson correlation and the low-loss proxy is not universally effective. However, the overall correlation between loss and cross-task performance is consistently positive, suggesting that low-loss prompts generally improve performance and, at worst, maintain the baseline without degradation.
> Empirically, SAPO achieves gains in almost all 72 experiments (Tables 1 and 9). While some metric improvements seem modest, they are averaged across multiple tasks, including weakly correlated ones, indicating SAPO achieves notable gains on some tasks. Furthermore, selecting worse prompts consistently degrades performance (Table 2), validating SAPO'seffectiveness.
> We attempt to analyze scenarios where the proxy fails from various angles (e.g., task type, difficulty, and gradients), but our analysis reveals no clear patterns among these weakly correlated tasks.
> In the revision, we will provide a more fine-grained breakdown of the performance changes brought by SAPO to further illustrate its effectiveness.

---

> > ### Author Rebuttal · Reviewer_EerM · 2026-04-03
> >
> > I would like to thank authors for the detailed rebuttal. Given that the rebuttal has addressed my concerns to some extent,  I will reconsider my score.

---

> > > ### Author Response · Authors · 2026-04-04
> > >
> > > We are very pleased that our response has addressed your concerns, and we sincerely thank you for your encouraging feedback and for your willingness to raise your score. Your comments have been highly valuable in improving the clarity and quality of our paper. We will carefully revise the manuscript following your suggestions and incorporate the additional experimental results and related discussion into the final version. We greatly appreciate the time and effort you have devoted to reviewing our work and are sincerely grateful for your support.

---

### Official Review · Reviewer_YK7h · 2026-03-13

**Soundness:** 3
**Presentation:** 3
**Significance:** 2
**Originality:** 3
**Overall Recommendation:** 4
**Confidence:** 3

**Summary:**

The paper's main claim is that semantically equivalent prompts can produce similar in-task performance while significantly different cross-task performance. Cross-task performance is measured by both forgetting of previously learned tasks and generalization to unseen tasks. These two cross-task effects are positively correlated, indicating that superior prompts exist despite semantic similarity. Based on this finding, the paper proposes SAPO, a lightweight training strategy that augments existing fine-tuning methods. SAPO selects the prompt with the lowest pre-update loss and demonstrates consistent performance gains over human-selected prompts across overall performance, forgetting, and generalization metrics.

**Compliance With Llm Reviewing Policy:**

Affirmed.

**Final Justification:**

the rebuttal clarified places I missed and addressed my concerns

**Key Questions For Authors:**

1. How sensitive is the superior prompt selection to hyperparameter choices during training?
2. Most experiments are run with only one or two seeds. How much variance exists across different initializations using the same prompts? In other words, how much of the observed gain is attributable to prompt selection versus random initialization differences?
3. Are there consistent traits shared among the superior prompts? For example, do they tend to be more detailed, more explicit about task constraints, or structurally different in some systematic way?

**Limitations:**

Yes

**Strengths And Weaknesses:**

## Strength:
1. The paper explores the impact of training prompts, which is an underexplored dimension of LLM fine-tuning. The controlled analysis of how training prompts affect cross-task performance is well-designed. The finding that semantically equivalent prompts can produce drastically different cross-task behaviors is surprising and interesting.

2. SAPO is easy to implement as a lightweight, plug-and-play method that is compatible with existing continual learning approaches, consistently bringing additional gains across baselines.

## Weakness:

1. My biggest concern is that the tasks selected are not sufficiently challenging or diverse given the models tested. The paper primarily reports gains as (S_post − S_original)/S_original, which obscures the absolute scale of S_original. It is unclear whether S_original represents a meaningful baseline from which to assess the practical significance of the improvements.

2. Relatedly, the SuperNI benchmark remains quite far from real-world deployment scenarios, which involve noisier objectives, more heterogeneous user requests, and longer-horizon tasks. It is unclear how well SAPO would generalize beyond controlled benchmark-style continual learning.

3. The paper does not make sufficient connection to the broader prompt optimization literature, such as DSPy and TextGrad.
SAPO selects prompts solely using the current-task pre-update loss, which rests on the unproven assumption that what is easiest to fit on the current task also minimizes forgetting and maximizes future-task generalization.

4. The paraphrase is only generated by Gemini 2.5. This dependency is not fully analyzed

---

> ### Author Rebuttal · Authors · 2026-03-30
>
> We sincerely thank the reviewer for the insightful feedback.
>
> **W1**: Our probe experiments use 26 diverse datasets from SuperNI, a widely adopted benchmark for continual learning [1, 2]. To concisely illustrate our settings, we categorize these into classification and generation tasks. Actually, as shown in Table 4, they cover a broad spectrum (e.g., question answering, summarization, program execution), ensuring a systematic and robust evaluation of prompt impacts.
> We also evaluate on TRACE benchmark, which introduces more complex tasks like mathematical reasoning and code generation.
>
> We report relative changes to highlight performance variance and normalize correlation analyses. Regarding $S_{original}$, Qwen3-8B's initial Rouge-L scores across 26 SuperNI tasks span from 5 to 90. For instance, the tasks in NI-Probe-C1, G1, and M1 exhibit initial scores of (58.9, 81.4, 54.8), (12.9, 64.2, 5.1), and (81.4, 22.2, 61.8), confirming a mix of straightforward and challenging tasks. Additionally, Section 5 reports absolute performance metrics, where our method demonstrates consistent improvements.
> In the revision, we will detail the information to better illustrate the practical significance of our improvements.
>
> **W2**: SuperNI encompasses diverse tasks, providing an ideal setting to systematically investigate the impact of training prompts. And it is widely utilized to demonstrate effectiveness of recent continual learning methods [1, 2]. Additionally, we also evaluate on TRACE, which incorporates more heterogeneous reasoning tasks.
>
> **W3**: As clarified in related work, existing prompt optimization methods focus on improving performance without training. In contrast, SAPO optimizes prompts to enhance post-training cross-task capabilities.
> Crucially, selecting low-loss prompts is a rigorous empirical finding, not an unproven assumption. Our systematic evaluation across 120 task sequences reveals that pre-update loss has the highest positive correlation with post-training performance. We further validate in Section 5.4 that low-loss prompts successfully minimize inter-task optimization conflicts.
> Metrics like Rouge-L (often targeted by existing prompt optimization methods) show weak correlation. And for models like Qwen3-8B, simple prompt paraphrasing induces an absolute Rouge-L variance of over 10 points on most tasks, rivaling the gains of prompt optimization methods.
> We will further clarify these distinctions and connections in the revision.
>
> **W4**: Our work reveals that even semantically consistent paraphrases of training prompts significantly impact performance, and SAPO effectively identifies the optimal variant.
> Generating such simple, diverse variations is trivial for modern LLMs.
> The table below reports averaged SAPO results with 3 runs on Qwen3-8B and NI-Seq-M1 sequence, using Gemini-2.5-Pro, GPT-OSS-120B, and Qwen3-32B as paraphrasers. SAPO achieves consistent improvements, demonstrating it does not rely on Gemini. We will add a more comprehensive analysis to the revision.
> |Baseline|Paraphraser|Δ AP|Δ BWT|Δ FWT|
> |:---|:---|:---|:---|:---|
> |LoraInc|Gemini|+1.89|+0.50|+0.06|
> ||GPT|+1.47|+1.21|+0.31|
> ||Qwen|+1.66|+0.62|+0.05|
> |O-Lora|Gemini|+1.07|+0.53|+2.97|
> ||GPT|+1.11|+0.47|+1.79|
> ||Qwen|+0.55|+0.76|+2.49|
>
> **Q1**: We provide a hyperparameter sensitivity analysis in Appendix D, focusing on candidate pool size. A pool size of 20 yields consistent and stable performance gains, whereas smaller sizes result in inconsistent gains.
>
> **Q2**: The table below presents the average metrics and standard deviation (Mean ± SD) of Qwen3-8B on NI-Seq-M1 over 3 seeds. While random initializations introduce some variance, SAPO consistently achieves improvements whose magnitude clearly exceeds the standard deviation. Furthermore, SAPO consistently improves performance across 72 experiments (Tables 1 and 9), and selecting worse prompts leads to consistent performance degradation (Table 2), demonstrating the observed gains are attributable to superior prompt selection.
> In the revision, we will include the variance analysis to validate SAPO's effectiveness.
>
> ||AP| BWT|FWT|
> |:---|:---| :---|:---|
> |LoraInc|65.43±0.21|-2.08±0.15|4.15±0.18|
> |+SAPO|67.28±0.28|-1.55±0.21|4.23±0.16|
> |O-Lora|65.08±0.35|-1.58±0.18|-1.42±0.23|
> |+ SAPO|66.45±0.32|-0.05±0.22|1.52±0.12|
>
> **Q3**: The superior prompts are simply paraphrases of the original instruction. Surface-level metrics, such as length, structural complexity, or the explicitness of task constraints, do not exhibit obvious shared traits. We are actively investigating potential non-surface commonalities (e.g., latent representations or optimization dynamics). We will include more prompt examples in the revision to provide further insights.
>
> [1] Unlocking the power of function vectors for characterizing and mitigating catastrophic forgetting in continual instruction tuning. ICLR 2025
>
> [2] Recurrent Knowledge Identification and Fusion for Language Model Continual Learning. ACL 2025

---

> > ### Author Rebuttal · Reviewer_YK7h · 2026-04-03
> >
> > Thank you for the clarification and it addresses most of my concern.

---

> > > ### Author Response · Authors · 2026-04-04
> > >
> > > We are very pleased that our response has addressed your concerns, and we sincerely thank you for your encouraging feedback and for your willingness to raise your score. Your thoughtful comments and suggestions have been extremely helpful in improving the quality and clarity of our paper. Following your suggestions, we will carefully revise the paper and incorporate the additional experimental results and related discussion into the final version. We greatly appreciate the time and effort you have devoted to reviewing our work, and we are sincerely grateful for your valuable feedback and encouragement.

---

### Decision · Program_Chairs · 2026-04-30

**Decision:**

Accept (regular)

**Comment:**

This paper studies whether small variations in training prompts can meaningfully affect cross-task behavior during fine-tuning, especially forgetting and forward generalization. Reviewers found the empirical observation interesting, and method simple and lightweight. Overall, this is a sound and worthwhile paper with a clear empirical contribution, but also one whose claims should be interpreted in line with the current evaluation scope.

The main strength of the paper is that it highlights training prompt formulation as a factor that matters for post-training behavior, rather than treating semantically similar prompts as interchangeable. The rebuttal helped clarify points about robustness, including sensitivity to paraphrasers, seed variance, etc. Three reviewers indicated that their concerns were largely resolved after rebuttal.

The remaining concern is about scope. The evidence is still concentrated in a continual learning setup, with additional results on TRACE. While this is a reasonable controlled setting for studying forgetting and generalization, it does limit how broadly the conclusions can be interpreted. There is also some lingering ambiguity around how semantically equivalent the prompt variants really are in all cases, and the mechanistic explanation is still preliminary.